# Distribution-free binary classification: prediction sets, confidence intervals and calibration

**Chirag Gupta**[*1], **Aleksandr Podkopaev**[*1,2], **Aaditya Ramdas**[1,2]
Machine Learning Department[1]
Department of Statistics and Data Science[2]
Carnegie Mellon University
{chiragg,podkopaev,aramdas}@cmu.edu

## Abstract

We study three notions of uncertainty quantification—calibration, confidence intervals and prediction sets—for binary classification in the distribution-free setting, that is without making any distributional assumptions on the data. With a focus towards calibration, we establish a 'tripod' of theorems that connect these three notions for score-based classifiers. A direct implication is that distribution-free calibration is only possible, even asymptotically, using a scoring function whose level sets partition the feature space into at most countably many sets. Parametric calibration schemes such as variants of Platt scaling do not satisfy this requirement, while nonparametric schemes based on binning do. To close the loop, we derive distribution-free confidence intervals for binned probabilities for both fixed-width and uniform-mass binning. As a consequence of our 'tripod' theorems, these confidence intervals for binned probabilities lead to distribution-free calibration. We also derive extensions to settings with streaming data and covariate shift.

## 1 Introduction

Let $\mathcal{X}$ and $\mathcal{Y} = \{0, 1\}$ denote the feature and label spaces for binary classification. Consider a predictor $f : \mathcal{X} \to \mathcal{Z}$ that produces a prediction in some space $\mathcal{Z}$. If $\mathcal{Z} = \{0, 1\}$, $f$ corresponds to a point prediction for the class label, but often class predictions are based on a 'scoring function'. Examples are, $\mathcal{Z} = \mathbb{R}$ for SVMs, and $\mathcal{Z} = [0, 1]$ for logistic regression, random forests with class probabilities, or deep models with a softmax top layer. In such cases, a higher value of $f(X)$ is often interpreted as higher belief that $Y = 1$. In particular, if $\mathcal{Z} = [0, 1]$, it is tempting to interpret $f(X)$ as a probability, and hope that

$$f(X) \approx \mathbb{P}(Y = 1 \mid X). \tag{1}$$

However, such hope is unfounded, and in general (1) will be far from true without strong distributional assumptions, which may not hold in practice. Valid uncertainty estimates that are related to (1) can be provided, but ML models do not satisfy these out of the box. This paper discusses three notions of uncertainty quantification: calibration, prediction sets (PS) and confidence intervals (CI), defined next. A function $f : \mathcal{X} \to [0, 1]$ is said to be (perfectly) calibrated if

$$\mathbb{E}\left[Y \mid f(X) = a\right] = a \quad \text{a.s. for all } a \text{ in the range of } f. \tag{2}$$

Define $\mathcal{L} \equiv \{\{0\}, \{1\}, \{0, 1\}, \varnothing\}$ and fix $\alpha \in (0, 1)$. A function $S : \mathcal{X} \to \mathcal{L}$ is a $(1 - \alpha)$-PS if

$$\mathbb{P}(Y \in S(X)) \geqslant 1 - \alpha. \tag{3}$$

Finally, let $\mathcal{I}$ denote the set of all subintervals of $[0, 1]$. A function $C : \mathcal{X} \to \mathcal{I}$ is a $(1 - \alpha)$-CI if

$$\mathbb{P}(\mathbb{E}\left[Y \mid X\right] \in C(X)) \geqslant 1 - \alpha. \tag{4}$$

---

All three notions are 'natural' in their own sense, but also different at first sight. We show that they are in fact tightly connected (see Figure 1), and focus on the implications of this result for calibration. Our analysis is in the distribution-free setting, that is, we are concerned with understanding what kinds of valid uncertainty quantification is possible without distributional assumptions on the data.

Our work primarily extends the ideas of Vovk et al. [47, Section 5] and Barber [3]. We also discuss Platt scaling [36], binning [51] and the recent work of Vaicenavicius et al. [44]. Other related work is cited as needed, and further discussed in Section 5. All proofs appear ordered in the Appendix.

**Notation:** Let $P$ denote any distribution over $\mathcal{X} \times \mathcal{Y}$. In practice, the available labeled data is often split randomly into the *training set* and the *calibration set*. Typically, we use $n$ to denote the number of calibration data points, so $\{(X_i, Y_i)\}_{i \in [n]}$ is the calibration data, where we use the shorthand $[a] := \{1, 2, \ldots a\}$. A prototypical test point is denoted $(X_{n+1}, Y_{n+1})$. All data are drawn i.i.d. from $P$, denoted succinctly as $\{(X_i, Y_i)\}_{i \in [n+1]} \sim P^{n+1}$. As above, random variables are denoted in upper case. The learner observes realized values of all random variables $(X_i, Y_i)$, except $Y_{n+1}$. (All sets and functions are implicitly assumed to be measurable.)

## 2 Calibration, confidence intervals and prediction sets

Calibration captures the intuition of (1) but is a weaker requirement, and was first studied in the meteorological literature for assessing probabilistic rain forecasts [5, 7, 31, 39]. Murphy and Epstein [31] described the ideal notion of calibration, called *perfect calibration* (2), which has also been referred to as *calibration in the small* [45], or sometimes simply as *calibration* [7, 12, 44]. The types of functions that can achieve perfect calibration can be succinctly captured as follows.

**Proposition 1.** *A function $f : \mathcal{X} \to [0, 1]$ is perfectly calibrated if and only if there exists a space $\mathcal{Z}$ and a function $g : \mathcal{X} \to \mathcal{Z}$, such that*

$$f(x) = \mathbb{E}\left[Y \mid g(X) = g(x)\right] \quad \text{almost surely } P_X. \tag{5}$$

Vaicenavicius et al. [44] stated and gave a short proof for the 'only if' direction. While the other direction is also straightforward, together they lead to an appealingly simple and complete characterization. The proof of Proposition 1 is in Appendix A.

It is helpful to consider two extreme cases of Proposition 1. First, setting $g$ to be the identity function yields that the Bayes classifier $\mathbb{E}[Y|X]$ is perfectly calibrated. Second, setting $g(\cdot)$ to any constant implies that $\mathbb{E}[Y]$ is also a perfect calibrator. Naturally, we cannot hope to estimate the Bayes classifier without assumptions, but even the simplest calibrator $\mathbb{E}[Y]$ can only be approximated in finite samples. Since Proposition 1 states that calibration is possible iff the RHS of (5) is known exactly for some $g$, perfect calibration is impossible in practice. Thus we resort to satisfying the requirement (2) approximately, which is implicitly the goal of many empirical calibration techniques.

**Definition 1** (Approximate calibration). A predictor $f : \mathcal{X} \to [0, 1]$ is $(\varepsilon, \alpha)$-approximately calibrated for some $\alpha \in (0, 1)$ and a function $\varepsilon : [0, 1] \to [0, 1]$ if with probability at least $1 - \alpha$,

$$\left|\mathbb{E}\left[Y|f(X)\right] - f(X)\right| \leqslant \varepsilon(f(X)). \tag{6}$$

Note that when the definition is applied to a test point $(X_{n+1}, Y_{n+1})$, there are two sources of randomness: the randomness in $(X_{n+1}, Y_{n+1})$, and the randomness in $f, \varepsilon$. The randomness in $f, \varepsilon$ can be statistical (through random training data) or algorithmic (if the learning procedure is non-deterministic). All probabilistic statements in this paper should be viewed through this lens. Often in this paper, $\varepsilon$ is a constant function. In this case, we represent its value simply as $\varepsilon$. With increasing data, a good calibration procedure should have $\varepsilon$ vanishing to 0. We formalize this next.

**Definition 2** (Asymptotic calibration). A sequence of predictors $\{f_n\}_{n \in \mathbb{N}}$ from $\mathcal{X} \to [0, 1]$ is asymptotically calibrated at level $\alpha \in (0, 1)$ if there exists a sequence of functions $\{\varepsilon_n\}_{n \in \mathbb{N}}$ such that $f_n$ is $(\varepsilon_n, \alpha)$-approximately calibrated for every $n$, and $\varepsilon_n(f_n(X_{n+1})) = o_P(1)$.

For readers familiar with the $\ell_1$-ECE metric for calibration [32], we note that the $\ell_1$-ECE $\mathbb{E}_X |\mathbb{E}[Y|f(X)] - f(X)|$ is closely related to (6). For instance, when $\varepsilon$ is a constant function, (6) implies that with probability at least $1 - \alpha$ (over $\varepsilon, f$), $\mathbb{E}_X |\mathbb{E}[Y|f(X)] - f(X)| \leqslant (1-\alpha)\varepsilon + \alpha \leqslant \varepsilon + \alpha$. (Some readers may be more familiar with the definition of $\ell_1$-ECE for vector-valued predictions in multiclass problems. The ECE of the vectorized prediction $[1 - f, f]$ is also related to (6) since $\mathbb{E}_X |\mathbb{E}[Y|f(X)] - f(X)| = 0.5 \cdot \ell_1\text{-ECE}([1 - f, f])$.)

We show that the notions of approximate and asymptotic calibration are related to prediction sets (3) and confidence intervals (4). PSs and CIs are only 'informative' if the sets or intervals produced by

them are small: confidence intervals are measured by their length (denoted as $|C(\cdot)|$), and prediction sets are measured by their diameter ($\mathrm{diam}(S(\cdot)) := |\text{convex hull}(S(\cdot))|$). Observe that for binary classification, the diameter of a PS is either $0$ or $1$.

For a given distribution, one might expect prediction sets to have a larger diameter than the length of the confidence intervals, since we want to cover the actual value of $Y_{n+1}$ and not its (conditional) expectation. As an example, if $\mathbb{E}\left[Y|X=x\right] = 0.5$ for every $x$, then the shortest possible confidence interval is $(0.5, 0.5]$ whose diameter is $0$. However, a valid $(1-\alpha)$-PS has no choice but to output $\{0, 1\}$ for at least $(1-2\alpha)$ fraction of the points (and a random guess for the other $2\alpha$ fraction), and thus must have expected diameter $\geqslant 1 - 2\alpha$ even in the limit of infinite data.

Recently, Barber [3] built on an earlier result of Vovk et al. [47] to show that if an algorithm provides an interval $C$ which is a $(1-\alpha)$-CI for all product distributions $P^{n+1}$ (of the training data and test-point), then $S := C \cap \{0, 1\}$ is also a $(1-\alpha)$-PS whenever $P_X$ is a nonatomic distribution. Since this implication holds for all nonatomic distributions $P_X$, including the ones with $\mathbb{E}\left[Y|X\right] \equiv 0.5$ discussed above, it implies that distribution-free CIs must necessarily be wide. Specifically, their length cannot shrink to $0$ as $n \to \infty$. This can be treated as an impossibility result for the existence of (distribution-free) informative CIs.

One way to circumvent these impossibilities is to consider CIs for functions with 'lower resolution' than $\mathbb{E}\left[Y|X\right]$. To this end, we introduce a notion of a CI or PS 'with respect to $f$' (w.r.t. $f$). As we discuss in Section 3 (and Section 3.1 in particular), these notions are connected to calibration.

**Definition 3** (CI or PS w.r.t. $f$). A function $C : \mathcal{Z} \to \mathcal{I}$ is a $(1-\alpha)$-CI with respect to $f : \mathcal{X} \to \mathcal{Z}$ if
$$\mathbb{P}(\mathbb{E}\left[Y \mid f(X)\right] \in C(f(X))) \geqslant 1 - \alpha. \tag{7}$$
Analogously, a function $S : \mathcal{Z} \to \mathcal{L}$ is a $(1-\alpha)$-PS with respect to $f : \mathcal{X} \to \mathcal{Z}$ if
$$\mathbb{P}(Y \in S(f(X))) \geqslant 1 - \alpha. \tag{8}$$

When instantiated for a test point $(X_{n+1}, Y_{n+1})$, the probability in definitions (7) and (8) is not only over the test point, but also over the randomness in the pair $(f, C)$ or $(f, S)$, which are usually learned on labeled data. In order to produce PSs and CIs, one typically fixes a function $f$ learned on an independent split of the labeled data, and considers learning a $C$ or $S$ that provides guarantees (7) and (8). For example, $S$ can be produced using inductive conformal techniques [26, 34, 37]. In this case, $C$ or $S$ would be random as well; to make this explicit, we often denote $C$ or $S$ as $\widehat{C}_n$ or $\widehat{S}_n$.

# 3 Relating notions of distribution-free uncertainty quantification

As preluded to above, we consider a standard setting for valid distribution-free uncertainty quantification where the 'training' data is used to learn a scoring function $f : \mathcal{X} \to \mathcal{Z}$ and then held-out 'calibration' data is used to estimate uncertainty. We establish that in this setting, the notions of calibration, PSs and CIs are closely related. Figure 1 summarizes this section's takeaway message. Here, and in the rest of the section, if $P$ is the distribution of data, then we denote the distribution of the random variable $Z = f(X)$ as $P_{f(X)}$.

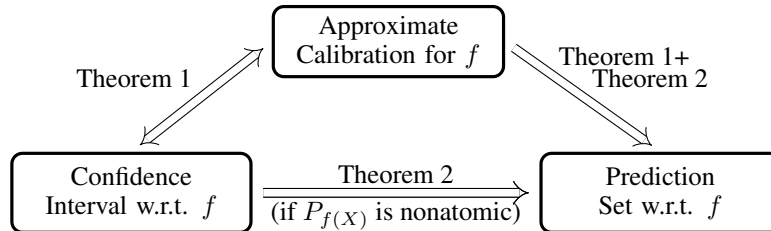

Figure 1: Relationship between notions of distribution-free uncertainty quantification.

In Section 3.1, we show that if an algorithm provides a CI, it can be used to provide a calibration guarantee and vice-versa (Theorem 1). In Section 3.2, we show that for all distributions $P$ such that $P_{f(X)}$ is nonatomic, if an algorithm constructs a distribution-free CI with respect to $f$, then it can be used to construct a distribution-free PS with respect to $f$ (Theorem 2). However, one typically expects the length of CIs to shrink to $0$ in the limit of infinite data, whereas PSs have a fixed distribution-dependent lower bound on their diameter. In Section 3.3, we use this intuition to prove a formal impossibility result for asymptotic calibration (Theorem 3). This result shows that for a

large class of standard scoring functions $f$ (such as logistic regression, deep networks with a final softmax layer, SVMs), it is impossible to achieve distribution-free asymptotic calibration without a 'discretization' step. Parametric schemes such as Platt scaling [36] do not perform such discretization and thus cannot lead to distribution-free calibration. To complement this lower bound, we provide calibration guarantees for one possible discretization step (histogram binning) in Section 4.

## 3.1 Relating calibration and confidence intervals

Given a predictor $f$ that is $(\varepsilon, \alpha)$-approximately calibrated, there is a trivial way to construct a function $C$ that is a $(1 - \alpha)$-CI: for $x \in \mathcal{X}$,

$$\underbrace{|\mathbb{E}\left[Y \mid f(x)\right] - f(x)| \leqslant \varepsilon(f(x))}_{\text{calibration}} \implies \underbrace{\mathbb{E}\left[Y \mid f(x)\right] \in C(f(x))}_{\text{CI w.r.t. } f} := [f(x) - \varepsilon(f(x)), f(x) + \varepsilon(f(x))]. \quad (9)$$

On the other hand, given $C$ that is a $(1 - \alpha)$-CI with respect to $f$, define for $z \in \mathrm{Range}(f)$ the left-endpoint, right-endpoint and midpoint functions respectively:

$$u_C(z) := \sup\left\{g : g \in C(z)\right\}, \; l_C(z) := \inf\left\{g : g \in C(z)\right\}, \; m_C(z) := (u_C(z) + l_C(z))/2. \quad (10)$$

Consider the midpoint $m_C(f(x))$ as a 'corrected' prediction for $x \in \mathcal{X}$:

$$\widetilde{f}(x) := m_C(f(x)), \; x \in \mathcal{X}, \quad (11)$$

and let $\varepsilon = \sup_{z \in \mathrm{Range}(f)} \left\{|C(z)|/2\right\}$ be a constant function returning the largest interval radius. Then $\widetilde{f}$ is $(\varepsilon, \alpha)$-approximately calibrated for a non-trivial $\varepsilon$. These claims are formalized next.

**Theorem 1.** *Fix any $\alpha \in (0, 1)$. Let $f : \mathcal{X} \to [0, 1]$ be a predictor that is $(\varepsilon, \alpha)$-approximately calibrated for some function $\varepsilon$. Then the function $C$ in (9) is a $(1 - \alpha)$-CI with respect to $f$.*

*Conversely, fix a scoring function $f : \mathcal{X} \to \mathcal{Z}$. If $C$ is a $(1 - \alpha)$-CI with respect to $f$, then the predictor $\widetilde{f}$ in (11) is $(\varepsilon, \alpha)$-approximately calibrated for $\varepsilon = \sup_{z \in Range(f)} \left\{|C(z)|/2\right\}$.*

The proof is in Appendix B. An important implication of Theorem 1 is that having a sequence of predictors that is asymptotically calibrated yields a sequence of confidence intervals with vanishing length as $n \to \infty$. This is formalized in the following corollary, also proved in Appendix B.

**Corollary 1.** *Fix any $\alpha \in (0, 1)$. If a sequence of predictors $\{f_n\}_{n \in \mathbb{N}}$ is asymptotically calibrated at level $\alpha$, then construction (9) yields a sequence of functions $\{C_n\}_{n \in \mathbb{N}}$ such that each $C_n$ is a $(1 - \alpha)$-CI with respect to $f_n$ and $|C_n(f_n(X_{n+1}))| = o_P(1)$.*

Next, we show that for a large class of scoring functions, CIs and PSs are also related in the distribution-free setting. This connection along with Corollary 2 (below) leads to an impossibility result for distribution-free asymptotic calibration for certain functions $f$ (Theorem 3 in Section 3.3).

## 3.2 Relating distribution-free confidence intervals and prediction sets

Suppose a function satisfies a CI guarantee with respect to $f$ no matter what the data-generating distribution $P$ is. We show that such a function would also provide a PS guarantee for all $P$ such that $P_{f(X)}$ is nonatomic. To write our theorem, we define the 'discretize' function to transform a confidence interval $C$ to a prediction set: $\mathrm{disc}(C) := C \cap \{0, 1\} \subseteq \mathcal{L}$. In the following theorem, the CI and PS guarantees provided (per equations (7) and (8)) are to be understood as marginal over both the calibration and test-data. To make this explicit, we denote the CI function as $\widehat{C}_n$.

**Theorem 2.** *Fix $\alpha \in (0, 1)$ and a function $f$ with domain $\mathcal{X}$. If $\widehat{C}_n$ is a $(1 - \alpha)$-CI with respect to $f$ for all distributions $P$, then $\mathrm{disc}(\widehat{C}_n)$ is a $(1 - \alpha)$-PS with respect to $f$ for all distributions $P$ for which $P_{f(X)}$ is nonatomic.*

The proof is in Appendix B. It adapts the proof of Barber [3, Theorem 1]. Their result connects the notions of CI and PS, but not with respect to $f$ (like in equations (3), (4)). By adapting the result for CIs and PSs with respect to $f$, and using Theorem 1, we are able to relate CIs and PSs to calibration and use this to prove an impossibility result for asymptotic calibration. This is done in the proof of Theorem 3 in the Section 3.3. A corollary of Theorem 2 that is used in Theorem 3 (but is also important on its own) is stated next.

**Corollary 2.** *Fix $f : \mathcal{X} \to \mathcal{Z}$ and $\alpha \in (0, 1)$. If $\widehat{C}_n$ is a $(1 - \alpha)$-CI with respect to $f$ for all $P$, and there exists a $P$ such that $P_{f(X)}$ is nonatomic, then we can construct a distribution $Q$ such that*

$$\mathbb{E}_{Q^{n+1}}|\widehat{C}_n(f(X_{n+1}))| \geqslant 0.5 - \alpha.$$

The proof is in Appendix B. For a given $f$, the bound in the corollary needs existence of $P$ such that $P_{f(X)}$ is nonatomic. These $f$ are characterized in the discussion after Corollary 3 (Section 3.3), and formally in the proof of Theorem 3. One expects the length of a confidence interval to vanish as $n \to \infty$. Corollary 2 shows that this is impossible in a distribution-free manner for certain $f$.

## 3.3 Necessary condition for distribution-free asymptotic calibration

Proposition 1 shows that a function $f$ is calibrated if and only if it takes the form (5) for some function $g$. Observe that in (5), the actual values taken by $g$ are only as informative as the *partition* of $\mathcal{X}$ induced by its level sets. Denote this partition as $\{\mathcal{X}_z\}_{z \in \mathcal{Z}}$, where $\mathcal{X}_z = \{x \in \mathcal{X} : g(x) = z\}$. Then we may equivalently define $f$ in (5) through a set of values $\{f_z = P(Y_{n+1} = 1 \mid X_{n+1} \in \mathcal{X}_z)\}_{z \in \mathcal{Z}}$, setting $f(\cdot) = f_{g(\cdot)}$. Thus we can re-interpret calibration as follows.

**Corollary 3** (to Proposition 1). *Any calibrated classifier $f$ is characterized by a partition of $\mathcal{X}$ into subsets $\{\mathcal{X}_z\}_{z \in \mathcal{Z}}$ and corresponding conditional probabilities $\{f_z\}_{z \in \mathcal{Z}}$ for some index set $\mathcal{Z}$.*

We now establish a necessary condition on the cardinality of the partition corresponding to an asymptotically calibrated scoring function. In the following statement, $\mathcal{X}^{(f)}$ denotes the level sets of $f$ on $\mathcal{X}$, and $|\mathcal{X}^{(f)}|$ denotes its cardinality (which may be infinite). Also $\aleph_0$ denotes the largest cardinality of a countable set, which is the cardinality of $\mathbb{N}$.

**Theorem 3.** *Let $\alpha \in (0, 0.5)$ be a fixed threshold. If a sequence of scoring functions $\{f_n\}_{n \in \mathbb{N}}$ is asymptotically calibrated at level $\alpha$ for every distribution $P$ then*

$$\limsup_{n \to \infty} |\mathcal{X}^{(f_n)}| \leqslant \aleph_0.$$

In words, the cardinality of the partition induced by $f_n$ must be at most countable for large enough $n$. The following phrasing is convenient: $f$ is said to lead to a *fine partition* of $\mathcal{X}$ if $|\mathcal{X}^{(f)}| > \aleph_0$. Then, for the purposes of distribution-free asymptotic calibration, Theorem 3 necessitates us to consider $f$ that do not lead to fine partitions. Popular scoring functions such as logistic regression, deep neural-nets with softmax output and SVMs lead to continuous $f$ that induce fine partitions of $\mathcal{X}$ and thus cannot be asymptotically calibrated without distributional assumptions.

The proof of Theorem 3 is in Appendix B, but we briefly sketch its intuition below. Corollary 1 shows that asymptotic calibration allows construction of CIs whose lengths vanish asymptotically. Corollary 2 shows however that asymptotically vanishing CIs are impossible (without distributional assumptions) for $f$ if there exists a distribution $P$ such that $P_{f(X)}$ is nonatomic. Consequently asymptotic calibration is also impossible for such $f$. If $\mathcal{Z} = \text{Range}(f)$ is countable, then by the axioms of probability, $\sum_{z \in \mathcal{Z}} \mathbb{P}(X \in \mathcal{X}_z) = \mathbb{P}(X \in \mathcal{X}) = 1$, and so $\mathbb{P}(X \in \mathcal{X}_z) \neq 0$ for at least some $z$. Thus $P_{f(X)}$ cannot be nonatomic for any $P$. On the other hand, if $\mathcal{Z}$ is uncountable we can show that there always exists a $P$ such that $P_{f(X)}$ is nonatomic. Hence distribution-free asymptotic calibration is impossible for such $f$, which leads to the result of the theorem.

Theorem 3 also applies to many parametric calibration schemes that 'recalibrate' an existing $f$ through a wrapper $h_n : \mathcal{Z} \to [0, 1]$ learnt on the calibration data, with the goal that $h_n(f(\cdot))$ is nearly calibrated: $\mathbb{E}\left[Y \mid h_n(f(X))\right] \approx h_n(f(X))$. For instance, consider methods like Platt scaling [36], temperature scaling [12] and beta calibration [20]. Each of these methods learns a continuous and monotonic[2] (hence bijective) wrapper $h_n$, and thus $\mathbb{E}\left[Y \mid h_n(f(X))\right] = \mathbb{E}\left[Y \mid f(X)\right]$. If $h_n$ is a good calibrator, we would have $\mathbb{E}\left[Y \mid f(X)\right] \approx h_n(f(X))$. One way to formalize this is to consider whether an interval around $h_n(f(X))$ is a CI for $\mathbb{E}\left[Y \mid f(X)\right]$. In other words — does there exist a function $\varepsilon_n : [0, 1] \to [0, 1]$ such that for every distribution $P$,

$$\widetilde{C}_n(f(X)) := [h_n(f(X)) - \varepsilon_n(h_n(f(X))), h_n(f(X)) + \varepsilon_n(h_n(f(X)))]$$

is a $(1 - \alpha)$-CI with respect to $f$ and $\varepsilon_n(h_n(f(X))) = o_P(1)$? Theorem 3 shows that this is impossible if $f$ leads to a fine partition of $\mathcal{X}$, irrespective of the properties of $h_n$. Thus the aforementioned parametric calibration methods cannot lead to asymptotic calibration in general (that is, without further distributional assumptions). It is likely that the implications of our result also applies to other continuous parametric methods that are not necessarily monotonic, as well as calibration schemes that aim to learn a calibrated predictor without sample-splitting.

On the other hand, the calibration methods of isotonic regression [52] and histogram binning [51] do provide a countable partition of $\mathcal{X}$, and thus may satisfy distribution-free approximate calibration guarantees. In Section 4, we analyze histogram binning and show that any scoring function can be 'binned' to achieve distribution-free calibration. We explicitly quantify the finite-sample approximate calibration guarantees that automatically also lead to asymptotic calibration. We also discuss calibration in the online setting and calibration under covariate shift.

# 4   Achieving distribution-free approximate calibration

In Section 4.1, we prove a distribution-free approximate calibration guarantee given a fixed partitioning of the feature space into finitely many sets. This calibration guarantee also leads to asymptotic calibration. In Section 4.2, we discuss a natural method for obtaining such a partition using sample-splitting, called histogram binning. Histogram binning inherits the bound in Section 4.1. This shows that binning schemes lead to distribution-free approximate calibration. In Section 4.3 and 4.4 we discuss extensions of this scheme to adaptive sampling and covariate shift respectively.

## 4.1   Distribution-free calibration given a fixed sample-space partition

Suppose we have a fixed partition of $\mathcal{X}$ into $B$ regions $\{\mathcal{X}_b\}_{b\in[B]}$, and let $\pi_b = \mathbb{E}\left[Y \mid X \in \mathcal{X}_b\right]$ be the expected label probability in region $\mathcal{X}_b$. Denote the partition-identity function as $\mathcal{B} : \mathcal{X} \to [B]$ where $\mathcal{B}(x) = b$ if and only if $x \in \mathcal{X}_b$. Given a calibration set $\{(X_i, Y_i)\}_{i\in[n]}$, let $N_b := |\{i \in [n] : \mathcal{B}(X_i) = b\}|$ be the number of points from the calibration set that belong to region $\mathcal{X}_b$. In this subsection, we assume that $N_b \geqslant 1$ (in Section 4.2 we show that the partition can be constructed to ensure that $N_b$ is $\Omega(n/B)$ with high probability). Define

$$\widehat{\pi}_b := \frac{1}{N_b} \sum_{i:\mathcal{B}(X_i)=b} Y_i \qquad \text{and} \qquad \widehat{V}_b := \frac{1}{N_b} \sum_{i:\mathcal{B}(X_i)=b} (Y_i - \widehat{\pi}_b)^2 \qquad (12)$$

as the empirical average and variance of the $Y$ values in a partition. We now deploy an empirical Bernstein bound [2] to produce a confidence interval for $\pi_b$.

**Theorem 4.** *For any $\alpha \in (0, 1)$, with probability at least $1 - \alpha$,*

$$|\pi_b - \widehat{\pi}_b| \leqslant \sqrt{\frac{2\widehat{V}_b \ln(3B/\alpha)}{N_b}} + \frac{3\ln(3B/\alpha)}{N_b}, \quad \textit{simultaneously for all } b \in [B].$$

The theorem is proved in Appendix C. Using the crude deterministic bound $\widehat{V}_b \leqslant 1$ we get that the length of the confidence interval for partition $b$ is $O(1/\sqrt{N_b})$. However, if for some $b$, $\mathcal{X}_b$ is highly informative or homogeneous in the sense that $\pi_b$ is close to 0 or 1, we expect $\widehat{V}_b \ll 1$. In this case, Theorem 4 *adapts* and provides an $O(1/N_b)$ length interval for $\pi_b$. Let $b^\star = \arg\min_{b\in[B]} N_b$ denote the index of the region with the minimum number of calibration examples.

**Corollary 4.** *For $\alpha \in (0, 1)$, the function $f_n(\cdot) := \widehat{\pi}_{\mathcal{B}(\cdot)}$ is $(\varepsilon, \alpha)$-approximately calibrated with*

$$\varepsilon = \sqrt{\frac{2\widehat{V}_{b^\star} \ln(3B/\alpha)}{N_{b^\star}}} + \frac{3\ln(3B/\alpha)}{N_{b^\star}}.$$

*Thus, $\{f_n\}_{n\in\mathbb{N}}$ is asymptotically calibrated at level $\alpha$.*

The proof is in Appendix C. Thus, any finite partition of $\mathcal{X}$ leads to asymptotic calibration. However, the finite sample guarantee of Corollary 4 can be unsatisfactory if the sample-space partition is chosen poorly, since it might lead to small $N_{b^\star}$. In Section 4.2, we present a data-dependent partitioning scheme that provably guarantees that $N_{b^\star}$ scales as $\Omega(n/B)$ with high probability. The calibration guarantee of Corollary 4 can also be stated conditional on a given test point $X_{n+1} = x$:

$$|\mathbb{E}\left[Y \mid f(X) = f(x)\right] - f(x)| \leqslant \varepsilon, \text{ almost surely } P_X . \qquad (13)$$

This holds since Theorem 4 provides simultaneously valid CIs for all regions $\mathcal{X}_b$.

## 4.2   Identifying a data-dependent partition using sample splitting

Here, we describe ways of constructing the partition $\{\mathcal{X}_b\}_{b\in[B]}$ through histogram binning [51], or simply, binning. Binning uses a sample splitting strategy to learn the partition of $\mathcal{X}$ as described in Section 4.1. A split of the data is used to learn the partition and an independent split is used

to estimate $\{\widehat{\pi}_b\}_{b \in [B]}$. Formally, the labeled data is split at random into a training set $\mathcal{D}_{\text{tr}}$ and a calibration set $\mathcal{D}_{\text{cal}}$. Then $\mathcal{D}_{\text{tr}}$ is used to train a scoring function $g : \mathcal{X} \to [0,1]$ (in general the range of $g$ could be any interval of $\mathbb{R}$ but for simplicity we describe it for $[0,1]$). The scoring function $g$ usually does not satisfy a calibration guarantee out-of-the-box but can be calibrated using binning.

A *binning scheme* $\mathcal{B}$ is any partition of $[0,1]$ into $B$ non-overlapping intervals $I_1, \ldots, I_B$, such that $\bigcup_{b \in [B]} I_b = [0,1]$ and $I_b \cap I_{b'} = \varnothing$ for $b \neq b'$. $\mathcal{B}$ and $g$ induce a partition of $\mathcal{X}$ as follows:

$$\mathcal{X}_b = \{x \in \mathcal{X} : g(x) \in I_b\}, \ b \in [B]. \tag{14}$$

The simplest binning scheme corresponds to *fixed-width binning*. In this case, bins have the form

$$I_i = \left[ \frac{i-1}{B}, \frac{i}{B} \right), i = 1, \ldots, B-1 \text{ and } I_B = \left[ \frac{B-1}{B}, 1 \right].$$

However, fixed-width binning suffers from the drawback that there may exist bins with very few calibration points (low $N_b$), while other bins may get many calibration points. For bins with low $N_b$, the $\widehat{\pi}_b$ estimates cannot be guaranteed to be well calibrated, since the bound of Theorem 4 could be large. To remedy this, we consider *uniform-mass binning*, which aims to guarantee that each region $\mathcal{X}_b$ contains approximately equal number of calibration points. This is done by estimating the empirical quantiles of $g(X)$. First, the calibration set $\mathcal{D}_{\text{cal}}$ is randomly split into two parts, $\mathcal{D}_{\text{cal}}^1$ and $\mathcal{D}_{\text{cal}}^2$. For $j \in [B-1]$, the $(j/B)$-th quantile of $g(X)$ is estimated from $\{g(X_i), i \in \mathcal{D}_{\text{cal}}^1\}$. Let us denote the empirical quantile estimates as $\widehat{q}_j$. Then, the bins are defined as:

$$I_1 = [0, \widehat{q}_1), I_i = [\widehat{q}_{i-1}, \widehat{q}_i), i = 2, \ldots, B-1 \text{ and } I_B = (\widehat{q}_{B-1}, 1].$$

This induces a partition of $\mathcal{X}$ as per (14). Now, only $\mathcal{D}_{\text{cal}}^2$ is used for calibrating the underlying classifier, as per the calibration scheme defined in Section 4.1. Kumar et al. [21] showed that uniform-mass binning provably controls the number of calibration samples that fall into each bin (see Appendix F.2). Building on their result and Corollary 4, we show the following guarantee.

**Theorem 5.** *Fix $g : \mathcal{X} \to [0,1]$ and $\alpha \in (0,1)$. There exists a universal constant $c$ such that if $|\mathcal{D}_{\text{cal}}^1| \geqslant cB \ln(2B/\alpha)$, then with probability at least $1 - \alpha$,*

$$N_{b^\star} \geqslant |\mathcal{D}_{\text{cal}}^2| / 2B - \sqrt{|\mathcal{D}_{\text{cal}}^2| \ln(2B/\alpha)/2}.$$

*Thus even if $|\mathcal{D}_{\text{cal}}^1|$ does not grow with $n$, as long as $|\mathcal{D}_{\text{cal}}^2| = \Omega(n)$, uniform-mass binning is $(\widetilde{O}(\sqrt{B \ln(1/\alpha)/n}), \alpha)$-approximately calibrated, and hence asymptotically calibrated for any $\alpha$.*

The proof is in Appendix C. In words, if we use a small number of points (independent of $n$) for uniform-mass binning, and the rest to estimate bin probabilities, we achieve approximate/asymptotic distribution-free calibration. Note that the probability is conditional on a fixed predictor $g$, and hence also conditional on the training data $\mathcal{D}_{\text{tr}}$. Since Theorem 5 uses Corollary 4, the calibration guarantee can also be stated conditionally on a fixed test point, akin to equation (13).

## 4.3 Distribution-free calibration in the online setting

So far, we have considered the batch setting with a fixed calibration set of size $n$. However, often a practitioner might want to query additional calibration data until a desired confidence level is achieved. This is called the *online* or *adaptive* setting. In this case, the results of Section 4 are no longer valid since the number of calibration samples is unknown a priori and may even be dependent on the data. In order to quantify uncertainty in the online setting, we use *time-uniform* concentration bounds [14, 15]; these hold simultaneously for all possible values of the calibration set size $n \in \mathbb{N}$.

Fix a partition of $\mathcal{X}$, $\{\mathcal{X}_b\}_{b \in [B]}$. For some value of $n$, let the calibration data be given as $\mathcal{D}_{\text{cal}}^{(n)}$. We use the superscript notation to emphasize the dependence on the current size of the calibration set. Let $\{(X_i^b, Y_i^b)\}_{i \in [N_b^{(n)}]}$ be examples from the calibration set that fall into the partition $\mathcal{X}_b$, where $N_b^{(n)} := |\{i \in [n] : \mathcal{B}(X_i) = b\}|$ is the total number of points that are mapped to $\mathcal{X}_b$. Let the empirical label average and cumulative (unnormalized) empirical variance be denoted as

$$\widehat{V}_b^+ = 1 \vee \sum_{i=1}^{N_b^{(n)}} \left( Y_i^b - \overline{Y}_{i-1}^b \right)^2, \text{ where } \overline{Y}_i^b := \frac{1}{i} \sum_{j=1}^{i} Y_j^b \text{ for } i \in [N_b^{(n)}]. \tag{15}$$

Note the normalization difference between $\widehat{V}_b^+$ and $\widehat{V}^b$ used in the batch setting. The following theorem constructs confidence intervals for $\{\pi_b\}_{b \in [B]}$ that are valid uniformly for any value of $n$.

**Theorem 6.** *For any $\alpha \in (0, 1)$, with probability at least $1 - \alpha$,*

$$|\pi_b - \widehat{\pi}_b| \leqslant \frac{7\sqrt{\widehat{V}_b^+ \ln\left(1 + \ln \widehat{V}_b^+\right)} + 5.3 \ln\left(\frac{6.3B}{\alpha}\right)}{N_b^{(n)}}, \quad \textit{simultaneously for all } b \in [B] \textit{ and all } n \in \mathbb{N}.$$

(16)

*Thus $\widehat{\pi}_b$ is asymptotically calibrated at any level $\alpha \in (0, 1)$.*

The proof is in Appendix C. Due to the crude bound: $\widehat{V}_b^+ \leqslant N_b^{(n)}$, we can see that the width of confidence intervals roughly scales as $O(\sqrt{\ln(1 + \ln N_b^{(n)})/N_b^{(n)}})$. In comparison to the batch setting, only a small price is paid for not knowing beforehand how many examples will be used for calibration.

## 4.4 Calibration under covariate shift

Here, we briefly consider the problem of calibration under covariate shift [41]. In this setting, calibration data $\{(X_i, Y_i)\}_{i \in [n]} \sim P^n$ is from a 'source' distribution $P$, while the test point is from a shifted 'target' distribution $(X_{n+1}, Y_{n+1}) \sim \widetilde{P} = \widetilde{P}_X \times P_{Y|X}$, meaning that the 'shift' occurs only in the covariate distribution while $P_{Y|X}$ does not change. We assume the likelihood ratio (LR)

$$w : \mathcal{X} \to \mathbb{R}; \quad w(x) := \mathrm{d}\widetilde{P}_X(x)/\mathrm{d}P_X(x)$$

is well-defined. The following is unambiguous: *if $w$ is arbitrarily ill-behaved and unknown, the covariate shift problem is hopeless, and one should not expect any distribution-free guarantees.* Nevertheless, one can still make nontrivial claims using a 'modular' approach towards assumptions:

Condition (A): $w(x)$ is known exactly and is bounded.
Condition (B): an asymptotically consistent estimator $\widehat{w}(x)$ for $w(x)$ can be constructed.

We show the following: under Condition (A), a weighted estimator using $w$ delivers approximate and asymptotic distribution-free calibration; under Condition (B), weighting with a plug-in estimator for $w$ continues to deliver asymptotic distribution-free calibration. It is clear that Condition (B) will always require distributional assumptions: asymptotic consistency is nontrivial for ill-behaved $w$. Nevertheless, the above two-step approach makes it clear where the burden of assumptions lie: not with calibration step, but with the $w$ estimation step. Estimation of $w$ is a well studied problem in the covariate-shift literature and there is some understanding of what assumptions are needed to accomplish it, but there has been less work on recognizing the resulting implications for calibration. Luckily, many practical methods exist for estimating $w$ given unlabeled samples from $\widetilde{P}_X$ [4, 16, 17]. In summary, if Condition (B) is possible, then distribution-free calibration is realizable, and if Condition (B) is not met (even with infinite samples), then it implies that $w$ is probably very ill-behaved, and so distribution-free calibration is also likely to be impossible.

For a fixed partition $\{\mathcal{X}_b\}_{b \in [B]}$, one can use the labeled data from the source distribution to estimate $\mathbb{E}_{\widetilde{P}}[Y \mid X \in \mathcal{X}_b]$ (unlike $\mathbb{E}_P[Y \mid X \in \mathcal{X}_b]$ as before), given oracle access to $w$:

$$\breve{\pi}_b^{(w)} := \frac{\sum_{i:\mathcal{B}(X_i)=b} w(X_i)Y_i}{\sum_{i:\mathcal{B}(X_i)=b} w(X_i)}.$$

(17)

As preluded to earlier, assume that

$$\text{for all } x \in \mathcal{X}, \ L \leqslant w(x) \leqslant U \text{ for some } 0 < L \leqslant 1 \leqslant U < \infty.$$

(18)

The 'standard' i.i.d. assumption on the test point equivalently assumes $w$ is known and $L = U = 1$. We now present our first claim: $\breve{\pi}_b^{(w)}$ satisfies a distribution-free approximate calibration guarantee. To show the result, we assume that the sample-space partition was constructed via uniform-mass binning (on the source domain) with sufficiently many points, as required by Theorem 5. This guarantees that all regions satisfy $|\{i : \mathcal{B}(X_i) = b\}| = \Omega(n/B)$ with high probability.

**Theorem 7.** *Assume $w$ is known and bounded* (18). *Then for an explicit universal constant $c > 0$, with probability at least $1 - \alpha$,*

$$\left|\breve{\pi}_b^{(w)} - \mathbb{E}_{\widetilde{P}}[Y \mid X \in \mathcal{X}_b]\right| \leqslant c\left(\frac{U}{L}\right)^2 \sqrt{\frac{B \ln(6B/\alpha)}{2n}}, \quad \textit{simultaneously for all } b \in [B],$$

*as long as $n \geqslant c(U/L)^2 B \ln^2(6B/\alpha)$. Thus $\breve{\pi}_b^{(w)}$ is asymptotically calibrated at any $\alpha \in (0, 1)$.*

The proof is in Appendix D. Theorem 7 establishes distribution-free calibration under Condition (A). For Condition (B), using $k$ *unlabeled* samples from the source and target domains, assume that we construct an estimator $\widehat{w}_k$ of $w$ that is consistent, meaning

$$\sup_{x \in \mathcal{X}} |\widehat{w}_k(x) - w(x)| \xrightarrow{P} 0. \tag{19}$$

We now define an estimator $\breve{\pi}_b^{(\widehat{w}_k)}$ by plugging in $\widehat{w}_k$ for $w$ in the right hand side of (17):

$$\breve{\pi}_b^{(\widehat{w}_k)} := \frac{\sum_{i:\mathcal{B}(X_i)=b} \widehat{w}_k(X_i) Y_i}{\sum_{i:\mathcal{B}(X_i)=b} \widehat{w}_k(X_i)}.$$

**Proposition 2.** *If $\widehat{w}_k$ is consistent* (19)*, then $\breve{\pi}_b^{(\widehat{w}_k)}$ is asymptotically calibrated at any $\alpha \in (0,1)$.*

In Appendix D, we illustrate through preliminary simulations that $w$ can be estimated using unlabeled data from the target distribution, and consequently approximate calibration can be achieved on the target domain. Recently, Park et al. [35] also considered calibration under covariate shift through importance weighting, but they do not show validity guarantees in the same sense as Theorem 7. For real-valued regression, distribution-free prediction sets under covariate shift were constructed using conformal prediction [42] under Condition (A), and is thus a precursor to our modular approach.

## 5  Other related work

The problem of assessing the calibration of binary classifiers was first studied in the meteorological and statistics literature [5–7, 9, 10, 28–31, 39]; we refer the reader to the review by Dawid [8] for more details. These works resulted in two common ways of measuring calibration: reliability diagrams [9] and estimates of the squared expected calibration error (ECE) [39]: $\mathbb{E}(f(X) - \mathbb{E}[Y \mid f(X)])^2$. Squared ECE can easily be generalized to multiclass settings and some related notions such as absolute deviation ECE and top-label ECE have also been considered, for instance [12, 32]. ECE is typically estimated through binning, which provably leads to underestimation of ECE for calibrators with continuous output [21, 44]. Certain methods have been proposed to estimate ECE without binning [50, 53], but they require distributional assumptions for provability.

While these papers have focused on the difficulty of *estimating* calibration error, ours is the first formal impossibility result for *achieving* calibration. In particular, Kumar et al. [21, Theorem 4.1] show that the scaling-binning procedure achieves calibration error close to the best within a fixed, regular, injective parametric class. However, as discussed in Section 3.3 (after Theorem 3), we show that the best predictor in an injective parametric class itself cannot have a distribution-free guarantee. In summary, our results show not only that (some form of) binning is necessary for distribution-free calibration (Theorem 3), but also sufficient (Corollary 4).

Apart from classical methods for calibration [33, 36, 51, 52], some new methods have been proposed recently, primarily for calibration of deep neural networks [12, 18, 19, 22, 23, 27, 40, 43, 49]. These calibration methods perform well in practice but do not have distribution-free guarantees. A calibration framework that generalizes binning and isotonic regression is Venn prediction [24, 45–48]; we briefly discuss this framework and show some connections to our work in Appendix E.

Calibration has natural applications in numerous sensitive domains where uncertainty estimation is desirable (healthcare, finance, forecasting). Recently, calibrated classifiers have been used as a part of the pipeline for anomaly detection [13, 25] and label shift estimation [1, 11, 38].

## 6  Conclusion

We analyze calibration for binary classification problems from the standpoint of robustness to distributional assumptions. By connecting calibration to other ways of quantifying uncertainty, we establish that popular parametric scaling methods cannot provide provable informative calibration guarantees in the distribution-free setting. In contrast, we showed that a standard nonparametric method — histogram binning — satisfies approximate and asymptotic calibration guarantees without distributional assumptions. We also establish guarantees for the cases of streaming data and covariate shift.

**Takeaway message.** Recent calibration methods that perform binning on top of parametric methods (Platt-binning [21] and IROvA-TS [53]) have achieved strong empirical performance. In light of the theoretical findings in our paper, we recommend some form of binning as the last step of calibrated prediction due to the robust distribution-free guarantees provided by Theorem 4.

# 7   Broader Impact

Machine learning is regularly deployed in real-world settings, including areas having high impact on individual lives such as granting of loans, pricing of insurance and diagnosis of medical conditions. Often, instead of hard $0/1$ classifications, these systems are required to produce soft probabilistic predictions, for example of the probability that a startup may go bankrupt in the next few years (in order to determine whether to give it a loan) or the probability that a person will recover from a disease (in order to price an insurance product). Unfortunately, even though classifiers produce numbers between 0 and 1, these are well known to not be 'calibrated' and hence not be interpreted as probabilities in any real sense, and using them in lieu of probabilities can be both misleading (to the bank granting the loan) and unfair (to the individual at the receiving end of the decision).

Thus, following early research in meteorology and statistics, in the last couple of decades the ML community has embraced the formal goal of calibration as a way to quantify uncertainty as well as to interpret classifier outputs. However, there exist other alternatives to quantify uncertainty, such as confidence intervals for the regression function and prediction sets for the binary label. There is not much guidance on which of these should be employed in practice, and what the relationship between them is, if any. Further, while there are many post-hoc calibration techniques, it is unclear which of these require distributional assumptions to work and which do not—this is critical because making distributional assumptions (for convenience) on financial or medical data is highly suspect.

This paper explicitly relates the three aforementioned notions of uncertainty quantification without making distributional assumptions, describes what is possible and what is not. Importantly, by providing distribution-free guarantees on well-known variants of binning, we identify a conceptually simple and theoretically rigorous way to ensure calibration in high-risk real-world settings. Our tools are thus likely to lead to fairer systems, better estimates of risks of high-stakes decisions, and more human-interpretable outputs of classifiers that apply out-of-the-box in many real-world settings because of the assumption-free guarantees.

## Acknowledgements

The authors would like to thank Tudor Manole, Charvi Rastogi, Michael Cooper Stanley, and the anonymous Neurips 2020 reviewers for comments on an initial version of this paper.

## Footnotes

[2]This assumes that the parameters satisfy natural constraints as discussed in the original papers: $a, b \geqslant 0$ for beta scaling with at least one of them nonzero, $A < 0$ for Platt scaling and $T > 0$ for temperature scaling.

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
