[Supplementary Material 1 · calibrated_prediction_full.pdf]

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

# Appendix

The Appendix contains proofs of results in the main paper ordered as they appear. Auxiliary results needed for some of the proofs are stated in Appendix F.

## A  Proof of Proposition 1

The 'if' part of the theorem is due to Vaicenavicius et al. [44, Proposition 1]; we reproduce it for completeness. Let $\sigma(g), \sigma(f)$ be the sub $\sigma$-algebras generated by $g$ and $f$ respectively. By definition of $f$, we know that $f$ is $\sigma(g)$-measurable and, hence, $\sigma(f) \subseteq \sigma(g)$. We now have:

$$
\begin{aligned}
\mathbb{E}\left[Y \mid f(X)\right] &= \mathbb{E}\left[\mathbb{E}\left[Y \mid g(X)\right] \mid f(X)\right] &&\text{(by tower rule since } \sigma(f) \subseteq \sigma(g)) \\
&= \mathbb{E}\left[f(X) \mid f(X)\right] &&\text{(by property (5))} \\
&= f(X).
\end{aligned}
$$

The 'only if' part can be verified for $g = f$. Since $f$ is perfectly calibrated,

$$
\mathbb{E}\left[Y \mid f(X) = f(x)\right] = f(x),
$$

almost surely $P_X$.

$\square$

## B  Proofs of results in Section 3

### B.1  Proof of Theorem 1

Assume that one is given a predictor $f$ that is $(\varepsilon, \alpha)$-approximately calibrated. Then the assertion follows from the definition of $(\varepsilon, \alpha)$-approximate calibration since:

$$
\left|\mathbb{E}\left[Y \mid f(X)\right] - f(X)\right| \leqslant \varepsilon(f(X)) \implies \mathbb{E}\left[Y \mid f(X)\right] \in C(f(X)).
$$

Now we show the proof in the other direction. If $m_C$ was injective, $\mathbb{E}\left[Y \mid m_C(f(X))\right] = \mathbb{E}\left[Y \mid f(X)\right]$ and thus if $\mathbb{E}\left[Y \mid f(X)\right] \in C(f(X))$ (which happens with probability at least $1 - \alpha$), we would have $\mathbb{E}\left[Y \mid m_C(f(X))\right] \in C(f(X))$ and so

$$
\left|\mathbb{E}\left[Y \mid m_C(f(X))\right] - m_C(f(X))\right| \leqslant \sup_{z \in \text{Range}(f)} \{|C(z)|/2\} = \varepsilon.
$$

This serves as an intuition for the proof in the general case, when $m_C$ need not be injective. Note that,

$$
\begin{aligned}
\left|\mathbb{E}\left[Y \mid m_C(f(X))\right] - m_C(f(X))\right| &= \left|\mathbb{E}\left[Y \mid m_C(f(X))\right] - \mathbb{E}\left[m_C(f(X)) \mid m_C(f(X))\right]\right| \\
&\stackrel{(1)}{=} \left|\mathbb{E}\left[\mathbb{E}\left[Y \mid f(X)\right] \mid m_C(f(X))\right] - \mathbb{E}\left[m_C(f(X)) \mid m_C(f(X))\right]\right| \\
&\stackrel{(2)}{=} \left|\mathbb{E}\left[\mathbb{E}\left[Y \mid f(X)\right] - m_C(f(X)) \mid m_C(f(X))\right]\right| \\
&\stackrel{(3)}{\leqslant} \mathbb{E}\left[\left|\mathbb{E}\left[Y \mid f(X)\right] - m_C(f(X))\right| \mid m_C(f(X))\right], \quad\quad (20)
\end{aligned}
$$

where we use the tower rule in (1) (since $m_C$ is a function of $f$), linearity of expectation in (2) and Jensen's inequality in (3). To be clear, the outermost expectation above is over $f(X)$ (conditioned on $m_C(f(X))$). Consider the event

$$
A : \mathbb{E}\left[Y \mid f(X)\right] \in C(f(X)).
$$

On $A$, by definition we have:

$$
\left|\mathbb{E}\left[Y \mid f(X)\right] - m_C(f(X))\right| = \frac{u_C(f(X)) - l_C(f(X))}{2} \leqslant \sup_{z \in \text{Range}(f)} \left(\frac{|C(z)|}{2}\right) = \varepsilon.
$$

By monotonicity property of conditional expectation, we also have that conditioned on $A$,

$$
\mathbb{E}\left[\left|\mathbb{E}\left[Y \mid f(X)\right] - m_C(f(X))\right| \mid m_C(f(X))\right] \leqslant \mathbb{E}\left[\varepsilon \mid m_C(f(X))\right] = \varepsilon,
$$

with probability 1. Thus by the relationship proved in the series of equations ending in (20), we have that conditioned on $A$, with probability 1,

$$|\mathbb{E}\left[Y \mid m_C(f(X))\right] - m_C(f(X))| \leqslant \varepsilon.$$

Since we are given that $C$ is a $(1-\alpha)$-CI with respect to $f$, $\mathbb{P}(A) \geqslant 1-\alpha$. For any event $B$, it holds that $\mathbb{P}(B) \geqslant \mathbb{P}(B|A)\mathbb{P}(A)$. Setting

$$B : |\mathbb{E}\left[Y \mid m_C(f(X))\right] - m_C(f(X))| \leqslant \varepsilon,$$

we obtain:

$$\mathbb{P}\left(|\mathbb{E}\left[Y \mid m_C(f(X))\right] - m_C(f(X))| \leqslant \varepsilon\right) \geqslant 1-\alpha.$$

Thus, we conclude that $m_C(f(\cdot))$ is $(\varepsilon, \alpha)$-approximately calibrated. $\qquad\square$

## B.2  Proof of Corollary 1

Let $\{f_n\}_{n\in\mathbb{N}}$ be asymptotically calibrated sequence with the corresponding sequence of functions $\{\varepsilon_n\}_{n\in\mathbb{N}}$ that satisfy $\varepsilon_n(f_n(X_{n+1})) = o_P(1)$. From Theorem 1, we can construct corresponding functions $C_n$ that are $(1-\alpha)$-CI with respect to $f_n$ and satisfy

$$|C_n(f_n(X_{n+1}))| = 2\varepsilon_n(f_n(X_{n+1})) = o_P(1).$$

This concludes the proof. $\qquad\square$

## B.3  Proof of Theorem 2

In the proof we write the test point as $(X_{n+1}, Y_{n+1})$. Suppose $\widehat{C}_n$ is a $(1-\alpha)$-CI with respect to $f$ for all distributions $P$. We show that $\widehat{C}_n$ covers the label $Y_{n+1}$ itself for distributions $P$ such that $P_{f(X)}$ is nonatomic (and thus disc$(\widehat{C}_n)$ would also cover the labels).

Let $P$ be any distribution such that $P_{f(X)}$ is nonatomic. Fix a set of $m \geqslant n+1$ samples from the distribution $P$ denoted as $\mathcal{T} = \{(A^{(j)}, B^{(j)})\}_{j\in[m]}$. Given $\mathcal{T}$, consider a distribution $Q$ corresponding to the following sampling procedure for $(X, Y) \sim Q$:

sample an index $j$ uniformly at random from $[m]$ and set $(X, Y) = (A^{(j)}, B^{(j)})$.

The distribution function for $Q$ is given by

$$m^{-1}\sum_{j=1}^{m}\delta_{(A^{(j)}, B^{(j)})}.$$

where $\delta_{(a,b)}$ denotes the points mass at $(a, b)$. Note that $Q$ is only defined conditional on $\mathcal{T}$. Observe the following facts about $Q$:

- supp$(Q) = \{(A^{(j)}, B^{(j)})\}_{j\in[m]}$.
- Consider any $(x, y) \in$ supp$(Q)$. Let $(x, y) = (A^{(j)}, B^{(j)})$ for some $j \in [m]$. Then

$$\mathbb{E}_Q\left[Y \mid f(X) = f(x)\right] = \mathbb{E}_Q\left[Y \mid f(X) = f(A^{(j)})\right]$$
$$\overset{\xi_1}{=} \mathbb{E}_Q\left[Y \mid X = A^{(j)}\right]$$
$$\overset{\xi_2}{=} B^{(j)} = y.$$

Above $\xi_1$ holds since $P_{f(X)}$ is nonatomic so that the $f(X^{(i)})$'s are unique almost surely. Note that $P_{f(X)}$ is nonatomic only if $P_X$ itself is nonatomic. Thus the $A^{(j)}$'s are unique almost surely, and $\xi_2$ follow. In other words, if $(X, Y) \sim Q$, then we have

$$Y = \mathbb{E}_Q\left[Y \mid f(X)\right]. \tag{21}$$

Suppose the data distribution was $Q$, that is $\{(X_i, Y_i)\}_{i \in [n+1]} \sim Q^{n+1}$. Define the event that the CI guarantee holds as

$$E_1 : \mathbb{E}\left[Y_{n+1} \mid f(X_{n+1})\right] \in \widehat{C}_n(f(X_{n+1})), \tag{22}$$

and the event that the PS guarantee holds as

$$E_2 : Y_{n+1} \in \widehat{C}_n(f(X_{n+1})). \tag{23}$$

Then due to (21), the events are exactly the same under $Q$:

$$E_1 \stackrel{Q}{\equiv} E_2. \tag{24}$$

In particular, this means

$$\mathbb{P}_{Q^{n+1}}(\mathbb{E}_Q\left[Y_{n+1} \mid f(X_{n+1})\right] \in \widehat{C}_n(f(X_{n+1}))) = \mathbb{P}_{Q^{n+1}}(Y_{n+1} \in \widehat{C}_n(f(X_{n+1}))). \tag{25}$$

If $\widehat{C}_n$ is a distribution-free CI, then $\mathbb{P}_{Q^{n+1}}(E_1) \geqslant 1 - \alpha$ and thus $\mathbb{P}_{Q^{n+1}}(E_2) \geqslant 1 - \alpha$. This shows that for $Q$, $\text{disc}(\widehat{C}_n)$ is a $(1 - \alpha)$-PI. Note that $Q$ corresponds to sampling *with replacement* from a fixed set $\mathcal{T}$ where each element is drawn with respect to $P$. Although $Q \neq P$, we expect that as $m \to \infty$ (while $n$ is fixed), $Q$ and $P$ coincide. This would prove the result for general $P$. To formalize this intuition, we describe a distribution which is close to $Q$ but corresponds to sampling *without replacement* from $\mathcal{T}$ instead.

For this, now suppose that $\{(X_i, Y_i)\}_{i \in [n+1]} \sim R^{n+1}$ where $R^{n+1}$ corresponds to sampling without replacement from $\mathcal{T}$. Formally, to draw from $R^{n+1}$, we first draw a surjective mapping $\lambda : [n+1] \to [m]$ as

$$\lambda \sim \text{Unif} \ (n\text{-sized ordered subsets of } [m]),$$

and set $(X_i, Y_i) = (A^{(\lambda(i))}, B^{(\lambda(i))})$ for $i \in [n + 1]$.

First we quantify precisely the intuition that as $m \to \infty$, $Q^{n+1}$ and $R^{n+1}$ are essentially identical. Consider the event $T :=$ no index is repeated in $Q^{n+1}$. Let $\mathbb{P}(T) = \tau_m$ for some $m$ and note that $\lim_{m \to \infty} \tau_m = 1$. Now consider any probability event $E$ over $\{(X_i, Y_i)\}_{i \in [n+1]}$ (such as $E_1$ or $E_2$). We have

$$\mathbb{P}_{Q^{n+1}}(E) = \mathbb{P}_{Q^{n+1}}(E|T) \cdot \mathbb{P}(T) + \mathbb{P}_{Q^{n+1}}(E|T^c) \cdot \mathbb{P}(T^c)$$
$$\in \left[\mathbb{P}_{Q^{n+1}}(E|T) \cdot \mathbb{P}(T), \mathbb{P}_{Q^{n+1}}(E|T) \cdot \mathbb{P}(T) + \mathbb{P}(T^c)\right].$$

Now observe that $\mathbb{P}_{Q^{n+1}}(E|T) = \mathbb{P}_{R^{n+1}}(E)$ to conclude

$$\mathbb{P}_{Q^{n+1}}(E) \in \left[\mathbb{P}_{R^{n+1}}(E) \cdot \mathbb{P}(T), \mathbb{P}_{R^{n+1}}(E) \cdot \mathbb{P}(T) + \mathbb{P}(T^c)\right].$$

Since $m \geqslant n + 1$, $\mathbb{P}(T) \neq 0$ so we can invert the above and substitute $\tau_m = \mathbb{P}(T)$ to get

$$\mathbb{P}_{R^{n+1}}(E) \in \left[\tau_m^{-1}(\mathbb{P}_{Q^{n+1}}(E) - (1 - \tau_m)), \ \tau_m^{-1}\mathbb{P}_{Q^{n+1}}(E)\right]. \tag{26}$$

Consider $E = E_2$ defined in equation (23). We showed that $\mathbb{P}_{Q^{n+1}}(E_2) \geqslant 1 - \alpha$. Thus from (26),

$$\mathbb{P}_{R^{n+1}}(E_2) \geqslant \tau_m^{-1}(1 - \alpha - (1 - \tau_m)).$$

The above is with respect to $R^{n+1}$ which is conditional on a fixed draw $\mathcal{T}$. However since the right hand side is independent of $\mathcal{T}$, we can also include the randomness in $\mathcal{T}$ to say:

$$\mathbb{P}_{R^{n+1}, \mathcal{T}}(E_2) \geqslant \tau_m^{-1}(1 - \alpha - (1 - \tau_m)). \tag{27}$$

Observe that if we consider the marginal distribution over $R^{n+1}$ and $\mathcal{T}$ (that is we include the randomness in $\mathcal{T}$ as above), $\{(X_i, Y_i)\}_{i \in [n+1]} \stackrel{iid}{\sim} P$. This is not true if we do not marginalize over $\mathcal{T}$, in particular since the $(X_i, Y_i)$'s are not independent (due to sampling without replacement). Thus equation (27) can be restated as

$$\mathbb{P}_{P^{n+1}}(E_2) \geqslant \tau_m^{-1}(1 - \alpha - (1 - \tau_m)),$$

Since $m$ can be set to any number and $\lim_{m \to \infty} \tau_m = 1$, we can indeed conclude

$$\mathbb{P}_{P^{n+1}}(E_2) \geqslant 1 - \alpha.$$

Recall that $E_2$ is the event that $Y_{n+1} \in \widehat{C}_n(X_{n+1})$; equivalently $Y_{n+1} \in \text{disc}\widehat{C}_n(X_{n+1})$. Thus $\text{disc}(\widehat{C}_n)$ provides a $(1 - \alpha)$-PI for $P$ such that $P_{f(X)}$ is nonatomic. $\qquad \square$

## B.4 Proof of Corollary 2

Let $P$ be any distribution such that $P_{f(X)}$ is nonatomic. By Theorem 2, $\widehat{C}_n$ must provide both a prediction set and a confidence interval for $P$:

$$\mathbb{P}(\mathbb{E}\left[Y_{n+1} \mid f(X_{n+1})\right] \in \widehat{C}_n(f(X_{n+1}))) \geqslant 1 - \alpha,$$

and

$$\mathbb{P}(Y_{n+1} \in \widehat{C}_n(f(X_{n+1}))) \geqslant 1 - \alpha.$$

Thus by a union bound

$$\mathbb{P}_{P^{n+1}}(\{Y_{n+1}, \mathbb{E}\left[Y_{n+1} \mid f(X_{n+1})\right]\} \subseteq \widehat{C}_n(f(X_{n+1}))) \geqslant 1 - 2\alpha. \tag{28}$$

Now consider a distribution $P$ such that $P_{f(X)}$ is nonatomic and $\mathbb{P}(Y = 1 \mid X) = 0.5$ a.s. $P_X$ so that $\mathbb{E}\left[Y_{n+1} \mid f(X)\right] = 0.5$ a.s. $P_{f(X)}$. The inequality (28) is true for this $P$ as well. If

$$\{Y_{n+1}, \mathbb{E}\left[Y_{n+1} \mid f(X_{n+1})\right]\} \subseteq \widehat{C}_n(f(X_{n+1})),$$

then $|\widehat{C}_n(X_{n+1})| \geqslant |Y_{n+1} - \mathbb{E}\left[Y_{n+1} \mid f(X_{n+1})\right]| \geqslant 0.5$. Thus

$$\mathbb{P}_{P^{n+1}}(|\widehat{C}_n(f(X_{n+1}))| \geqslant 0.5) \geqslant 1 - 2\alpha.$$

Consequently we have

$$\mathbb{E}_{P^{n+1}}|\widehat{C}_n(f(X_{n+1}))| \geqslant 0.5(1 - 2\alpha)$$
$$= 0.5 - \alpha.$$

This concludes the proof. $\qquad\square$

## B.5 Proof of Theorem 3

Suppose that $\{f_n\}_{n \in \mathbb{N}}$ is asymptotically calibrated and satisfies

$$\limsup_{n \to \infty} \left| \mathcal{X}^{(f_n)} \right| > \aleph_0,$$

that is, for every $m \in \mathbb{N}$, there exists $n \geqslant m$ such that $\mathcal{X}^{(f_n)}$ is an uncountable set. We will show a contradiction using Corollary 2 for $f_n$ and a certain $C_n$ to be defined shortly.

First, we verify the condition of Corollary 2 for $f_n$ if $\mathcal{X}^{(f_n)}$ is uncountable: we construct a distribution $P$ such that $P_{(f_n(X))}$ is nonatomic. Let the range of $f_n$ acting on $\mathcal{X}$ be denoted as $f_n(\mathcal{X})$, and for $z \in f_n(\mathcal{X})$ let the level set at value $z$ be denoted as $\mathcal{X}_z^{(f_n)}$. Since the sets $\mathcal{X}^{(f_n)}$ are measurable, we can define $P(X)$ as follows:

$$P(f_n(X)) = \text{Unif}(f_n(\mathcal{X})); \quad P(X \mid f_n(X)) = \text{Unif}\left(\mathcal{X}_{f_n(X)}^{(f_n)}\right). \tag{29}$$

$P(X)$ along with any conditional probability function $P(Y \mid X)$ constitutes a valid probability distribution $P$. Further, from the construction, since $\mathcal{X}^{(f_n)}$ is uncountable, $P_{f_n(X)}$ is guaranteed to be nonatomic.

Next, since $\{f_n\}_{n \in \mathbb{N}}$ is asymptotically calibrated, by Corollary 1, one can construct a sequence of functions $\{C_n\}_{n \in \mathbb{N}}$ such that each $C_n$ is a $(1 - \alpha)$-CI with respect to $f_n$ for any distribution $Q$, and

$$|C_n(f_n(X_{n+1}))| = o_Q(1).$$

Thus there exists a constant $m$ such that for $n \geqslant m$ and any distribution $Q$,

$$\mathbb{E}_{Q^{n+1}} |C_n(f_n(X_{n+1}))| < 0.5 - \alpha. \tag{30}$$

However, since $\limsup_{n \to \infty} |\mathcal{X}^{(f_n)}| > \aleph_0$, there exists an $n \geqslant m$ such that $\mathcal{X}^{(f_n)}$ is uncountable. Hence the requirements of Corollary 2 are satisfied by $\widehat{C}_n$ and $f_n$: namely $\widehat{C}_n$ is a $(1 - \alpha)$-CI with respect to $f$ for all distributions $P$, and there exists a $P$ such that $P_{f_n(X)}$ is nonatomic. Thus Corollary 2 yields that we can construct a distribution $Q$ such that

$$\mathbb{E}_{Q^{n+1}} |C_n(f_n(X_{n+1}))| \geqslant 0.5 - \alpha,$$

which is a contradiction to (30). Hence our hypothesis that $\limsup_{n \to \infty} |\mathcal{X}^{(f_n)}| > \aleph_0$ must be false, concluding the proof. $\qquad\square$

# C   Proofs of results in Section 4 (other than Section 4.4)

## C.1   Proof of Theorem 4

Let $E_{\mathcal{B}(x)}$ be the event that $(\mathcal{B}(X_1), \ldots, \mathcal{B}(X_n)) = (\mathcal{B}(x_1), \ldots, \mathcal{B}(x_n))$. On the event $E_{\mathcal{B}(x)}$, within each region $\mathcal{X}_b$, the number of point from the calibration set is known and the $Y_i$'s in each bin represent independent Bernoulli random variables that share the same mean $\pi_b = \mathbb{E}\left[Y \mid X \in \mathcal{X}_b\right]$. Consider any fixed region $\mathcal{X}_b$, $b \in [B]$. Using Theorem 10, we obtain that:

$$\mathbb{P}\left( |\pi_b - \widehat{\pi}_b| > \sqrt{\frac{2\widehat{V}_b \ln(3B/\alpha)}{N_b}} + \frac{3\ln(3B/\alpha)}{N_b} \;\middle|\; E_{\mathcal{B}(x)} \right) \leqslant \alpha/B.$$

Applying union bound across all regions of the sample-space partition, we get that:

$$\mathbb{P}\left( \forall b \in [B] : |\pi_b - \widehat{\pi}_b| \leqslant \sqrt{\frac{2\widehat{V}_b \ln(3B/\alpha)}{N_b}} + \frac{3\ln(3B/\alpha)}{N_b} \;\middle|\; E_{\mathcal{B}(x)} \right) \geqslant 1 - \alpha.$$

Because this is true for any $E_{\mathcal{B}(x)}$, we can marginalize to obtain the assertion of the theorem in unconditional form. $\qquad\square$

## C.2   Proof of Corollary 4

We show a calibration guarantee by using Theorem 1. Consider the scoring function as $\mathcal{B}$ with $\mathcal{Z} = [B]$. Then by Theorem 4, $C : [B] \to \mathcal{I}$ given by

$$C(b) = \left[ \widehat{\pi}_b - \left( \sqrt{\frac{2\widehat{V}_b \ln(3B/\alpha)}{N_b}} + \frac{3\ln(3B/\alpha)}{N_b} \right), \widehat{\pi}_b + \sqrt{\frac{2\widehat{V}_b \ln(3B/\alpha)}{N_b}} + \frac{3\ln(3B/\alpha)}{N_b} \right], \; b \in [B],$$

provides a $(1 - \alpha)$-CI with respect to $\mathcal{B}$. Let $b^\star = \min_{b \in [B]} N_b$. To apply Theorem 4, we define

$$\varepsilon = \sup_{b \in [B]} |C(b)| /2 = \sqrt{\frac{2\widehat{V}_{b^\star} \ln(3B/\alpha)}{N_{b^\star}}} + \frac{3\ln(3B/\alpha)}{N_{b^\star}},$$

and the mid-point function $m_C$ for $C$ is given by $m_C(b) = \widehat{\pi}_b$. Applying Theorem 1 gives the first part of the result.

Next, suppose some bin $b$ has $\mathbb{P}(\mathcal{B}(X) = b) = 0$. Then, a test point $X_{n+1}$ almost surely does not belong to the bin, and the bin can be ignored for our calibration guarantee. Thus without loss of generality, suppose every $b \in [B]$ satisfies

$$\mathbb{P}(\mathcal{B}(X) = b) > 0.$$

Let $\min_{b \in [B]} \mathbb{P}(\mathcal{B}(X) = b) = \tau > 0$. Then for a fixed number of samples $n$, any particular bin $b$, and any constant $\alpha \in (0, 1)$ we have by Hoeffding's inequality with probability $1 - \alpha/B$

$$N_b \geqslant n\tau - \sqrt{\frac{n \ln(B/\alpha)}{2}}.$$

Taking a union bound, we have with probability $1 - \alpha$, simultaneously for every $b \in [B]$,

$$N_b \geqslant n\tau - \sqrt{\frac{n \ln(B/\alpha)}{2}} = \Omega(n),$$

and in particular $N_{b^\star} = \Omega(n)$ where $b^\star = \arg\min_{b \in [B]} N_b$. Thus by the first part of this corollary, $f_n$ is $\varepsilon_n$ calibrated where $\varepsilon_n = O(\sqrt{n^{-1}}) = o(1)$. This concludes the proof. $\qquad\square$

## C.3 Proof of Theorem 5

Denote $|\mathcal{D}_{cal}^2| = n$. Let $p_j = \mathbb{P}(g(X) \in I_j)$ be the true probability that a random point falls into partition $\mathcal{X}_j$. Assume $c$ is such that we can use Lemma 11 to guarantee that with probability at least $1 - \alpha/2$, uniform mass binning scheme is 2-well-balanced. Hence, with probability at least $1 - \alpha/2$:

$$\frac{1}{2B} \leqslant p_j \leqslant \frac{2}{B}, \ \forall j \in [B]. \tag{31}$$

Moreover, by Hoeffding's inequality we get that for any fixed region of sample-space partition, with probability at least $1 - \alpha/2B$, for a fixed $j \in [B]$,

$$N_j \geqslant np_j - \sqrt{\frac{n \ln(2B/\alpha)}{2}}. \tag{32}$$

Hence, by union bound across applied accross all regions and using (31), we get that with probability at least $1 - \alpha/2$:

$$N_{b^\star} \geqslant \frac{n}{2B} - \sqrt{\frac{n \ln(2B/\alpha)}{2}},$$

where the first term dominates asymptotically (for fixed $B$). Hence, we get that with probability at least $1 - \alpha$, $N_{b^\star} = \Omega\left(n/B\right)$. By invoking the result of Corollary 4 and observing that $\widehat{V}_b \leqslant 1$, we conclude that uniform mass binning is $(\varepsilon, \alpha)$ approximately calibrated with $\varepsilon = O(\sqrt{B \ln(B/\alpha)/n})$ as desired. This also leads to asymptotic calibration by Corollary 4. $\qquad\square$

## C.4 Proof of Theorem 6

The proof is based on the result for an empirical-Bernstein confidence sequences for bounded observations [15]. We condition on the event $E_{\mathcal{B}(x)}^\infty$ defined as $(\mathcal{B}(X_1), \mathcal{B}(X_1), \dots) = (\mathcal{B}(x_1), \mathcal{B}(x_2), \dots)$, that is the random variables denoting which partition the infinite stream of samples fall in (thus allowing our bound to hold for every possible value of $n$). On $E_{\mathcal{B}(x)}^\infty$, the label values within each partition of the sample-space partition represent independent Bernoulli random variable that share the same mean $\pi_b = \mathbb{E}\left[Y \mid X \in \mathcal{X}_b\right], b \in [B]$. Consequently, the bound obtained can be marginalized over $E_{\mathcal{B}(x)}^\infty$ to obtain the assertion of the theorem in unconditional form. Now we show the bound that applies conditionally on $E_{\mathcal{B}(x)}^\infty$.

Consider any fixed region of the sample-space partition $\mathcal{X}_b$ and corresponding points $\{(X_i^b, Y_i^b)\}_{i=1}^{N_b}$. Then $S_t = \left(\sum_{i=1}^t Y_i^b\right) - t\pi_b$ is a sub-exponential process with variance process:

$$\widehat{V}_t^+ = \sum_{i=1}^t \left(Y_i^b - \overline{Y}_{i-1}^b\right)^2.$$

Howard et al. [14, Proposition 2] implies that $S_t$ is also a sub-gamma process with variance process $\widehat{V}_t$ and the same scale $c = 1$. Since the theorem holds for any sub-exponential uniform boundary, we choose one based on analytical convenience. Recall definition of the polynomial stitching function

$$\mathcal{S}_\alpha(v) := \sqrt{k_1^2 v l(v) + k_2^2 c^2 l^2(v)} + k_2 c l(v), \quad \text{where} \quad \begin{cases} l(v) := \ln h(\ln_\eta(v/m)) + \ln(l_0/\alpha), \\ k_1 := (\eta^{1/4} + \eta^{-1/4})/\sqrt{2}, \\ k_2 := (\sqrt{\eta} + 1)/\sqrt{2}. \end{cases}$$

where $l_0 = 1$ for the scalar case. Note that for $c > 0$ it holds that $\mathcal{S}_\alpha(v) \leqslant k_1 \sqrt{v l(v)} + 2 c k_2 l(v)$.

From Howard et al. [15, Theorem 1], it follows that $u(v) = \mathcal{S}_\alpha(v \vee m)$ is a sub-gamma uniform boundary with scale $c$ and crossing probability $\alpha$. Applying Theorem 9 with $h(k) \leftarrow (k+1)^s \zeta(s)$ where $\zeta(\cdot)$ is Riemann zeta function and parameters $\eta \leftarrow e$, $s \leftarrow 1.4$, $c \leftarrow 1$, $m \leftarrow 1$ and $\alpha \leftarrow \alpha/(2B)$, yields that $k_2 \leqslant 1.88, k_1 \leqslant 1.46$ and $l(v) = 1.4 \cdot \ln \ln (ev) + \ln(2\zeta(1.4)B/\alpha)$. Since Theorem 9 provides a bound that holds uniformly across time $t$, then it provides a guarantee for $t = N_b$, in particular. Hence, with probability at least $1 - \alpha/B$,

$$|\pi_b - \widehat{\pi}_b| \leqslant \frac{1.46\sqrt{\widehat{V}_b^+ \cdot 1.4 \cdot \ln \ln \left(e\left(\widehat{V}_b^+ \vee 1\right)\right) + \ln(6.3B/\alpha)}}{N_b} + \frac{5.27 \cdot \ln \ln \left(e\left(\widehat{V}_b^+ \vee 1\right)\right) + 3.76 \ln(6.3B/\alpha)}{N_b}$$

$$\leqslant \frac{7\sqrt{\widehat{V}_b^+ \cdot \ln \ln \left( e\left( \widehat{V}_b^+ \vee 1 \right) \right)} + 5.3\ln(6.3B/\alpha)}{N_b}.$$

using that $\sqrt{x + y} \leqslant \sqrt{x} + \sqrt{y}$ and $\ln \ln(ex) \leqslant \sqrt{x \ln \ln ex}$ for $x \geqslant 1$. Finally, we apply a union bound to get a guarantee that holds simultaneously for all regions of the sample-space partition. $\quad\square$

# D  Calibration under covariate shift (including proofs of results in Section 4.4)

The results from Section 4.4 are proved in Appendix D.1 (Theorem 7) and D.3 (Proposition 2). To show Theorem 7, we first propose and analyze a slightly different estimator than (39) that is unbiased for $\pi_b^{(w)}$, but needs additional oracle access to the parameters $\{m_b\}_{b \in [B]}$ defined as

$$m_b = \mathbb{P}_{P_X}(X \in \mathcal{X}_b) \,/\, \mathbb{P}_{\widetilde{P}_X}(X \in \mathcal{X}_b).$$

$m_b$ denotes the 'relative mass' of region $\mathcal{X}_b$. (For simplicity, we assume that $\mathbb{P}_{\widetilde{P}}(X \in \mathcal{X}_b) > 0$ for every $b$ since otherwise the test-point almost surely does not belong to $\mathcal{X}_b$ and estimation in that bin is not relevant for a calibration guarantee.) We then show that $m_b$ can be estimated using $w$, which would lead to the proposed estimator $\breve{\pi}_b^{(w)}$. First, we establish the following relationship between $\mathbb{E}_{\widetilde{P}}[Y \mid X \in \mathcal{X}_b]$ and $\mathbb{E}_P[Y \mid X \in \mathcal{X}_b]$.

**Proposition 3.** *Under the covariate shift assumption, for any $b \in [B]$*

$$\mathbb{E}_{\widetilde{P}}[Y \mid X \in \mathcal{X}_b] = m_b \cdot \mathbb{E}_P[w(X)Y \mid X \in \mathcal{X}_b].$$

*Proof.* Observe that

$$\frac{d\widetilde{P}(X \mid X \in \mathcal{X}_b)}{dP(X \mid X \in \mathcal{X}_b)} = \frac{d\widetilde{P}(X)}{dP(X)} \cdot \frac{\mathbb{P}_P(X \in \mathcal{X}_b)}{\mathbb{P}_{\widetilde{P}}(X \in \mathcal{X}_b)} = w(X) \cdot m_b.$$

Thus we have,

$$
\begin{aligned}
\mathbb{E}_{\widetilde{P}}[Y \mid X \in \mathcal{X}_b] &\stackrel{(1)}{=} \mathbb{E}_{\widetilde{P}}\left[ \mathbb{E}_{\widetilde{P}}[Y \mid X] \mid X \in \mathcal{X}_b \right] \\
&\stackrel{(2)}{=} \mathbb{E}_{\widetilde{P}}\left[ \mathbb{E}_P[Y \mid X] \mid X \in \mathcal{X}_b \right] \\
&\stackrel{(3)}{=} \mathbb{E}_P\left[ \frac{d\widetilde{P}(X \mid X \in \mathcal{X}_b)}{dP(X \mid X \in \mathcal{X}_b)} \cdot \mathbb{E}_P[Y \mid X] \mid X \in \mathcal{X}_b \right] \\
&\stackrel{(4)}{=} m_b \cdot \mathbb{E}_P\left[ w(X)\mathbb{E}_P[Y \mid X] \mid X \in \mathcal{X}_b \right] \\
&\stackrel{(5)}{=} m_b \cdot \mathbb{E}_P\left[ \mathbb{E}_P[w(X)Y \mid X] \mid X \in \mathcal{X}_b \right] \\
&\stackrel{(6)}{=} m_b \cdot \mathbb{E}_P\left[ w(X)Y \mid X \in \mathcal{X}_b \right],
\end{aligned}
$$

where in (1) we use the tower rule, in (2) we use the covariate shift assumption, (3) can be seen by using the integral form of the expectation, (4) uses the observation at the beginning of the proof, (5) uses that $w(X)$ is a function of $X$ and finally, (6) uses the tower rule. $\quad\square$

Let $N_b$ denote the number of calibration points from the source domain that belong to bin $b$. Given Proposition 3, a natural estimator for $\mathbb{E}_{\widetilde{P}}[Y \mid X \in \mathcal{X}_b]$ is given by:

$$\widehat{\pi}_b^{(w)} := \frac{1}{N_b} \sum_{i:\mathcal{B}(X_i)=b} m_b w(X_i)Y_i. \tag{33}$$

Estimation properties of $\widehat{\pi}_b^{(w)}$ are given by the following theorem.

**Theorem 8.** *Assume that* $\sup_x w(x) = U < \infty$. *For any* $\alpha \in (0,1)$, *with probability at least* $1 - \alpha$,

$$\left| \widehat{\pi}_b^{(w)} - \mathbb{E}_{\widetilde{P}}\left[ Y \mid X \in \mathcal{X}_b \right] \right| \leqslant \sqrt{\frac{2\widehat{V}_b^{(w)} \ln(3B/\alpha)}{N_b}} + \frac{3m_b U \ln(3B/\alpha)}{N_b}, \quad \textit{simultaneously for all } b \in [B],$$

*where* $\widehat{V}_b^{(w)} = \frac{1}{N_b} \sum_{i:\mathcal{B}(X_i)=b} (m_b w(X_i)Y_i - \widehat{\pi}_b^{(w)})^2$.

The proof is given in Appendix D.2. Next, we discuss a way of estimating $m_b$ using likelihood ratio $w$ instead of relying on oracle access. Observe that

$$\frac{d\widetilde{P}(X \mid X \in \mathcal{X}_b)}{dP(X \mid X \in \mathcal{X}_b)} = \frac{d\widetilde{P}(X)}{dP(X)} \cdot \frac{\mathbb{P}_P\left( X \in \mathcal{X}_b \right)}{\mathbb{P}_{\widetilde{P}}(X \in \mathcal{X}_b)} = w(X) \cdot m_b.$$

Thus we have,

$$\mathbb{E}_P\left[ w(X) \mid X \in \mathcal{X}_b \right] = m_b^{-1} \mathbb{E}_P \left[ \frac{d\widetilde{P}(X \mid X \in \mathcal{X}_b)}{dP(X \mid X \in \mathcal{X}_b)} \mid X \in \mathcal{X}_b \right] = m_b^{-1}, \qquad (34)$$

which suggests a possible estimator for $m_b$ given by

$$\widehat{m}_b = \left( \frac{\sum_{i:\mathcal{B}(X_i)=b} w(X_i)}{N_b} \right)^{-1}, \quad b \in [B]. \qquad (35)$$

On substituting this estimate for $m_b$ in (33), we get a new estimator

$$\frac{\sum_{i:\mathcal{B}(X_i)=b} w(X_i)Y_i}{\sum_{i:\mathcal{B}(X_i)=b} w(X_i)},$$

which is exactly $\breve{\pi}_b^{(w)}$. With this observation, we now prove Theorem 7.

### D.1 Proof of Theorem 7

Let us define $r_b := 1/m_b$ and

$$\widehat{r}_b = \frac{\sum_{i:\mathcal{B}(X_i)=b} w(X_i)}{N_b}. \qquad (36)$$

**Step 1 (Uniform lower bound for $N_b$).** Since the regions of the sample-space partition were constructed using uniform-mass binning, the guarantee of Theorem 5 holds. Precisely, we have that with probability at least $1 - \alpha/3$, simultaneously for every $b \in [B]$,

$$N_b \geqslant \frac{n}{2B} - \sqrt{\frac{n \ln(6B/\alpha)}{2}}.$$

**Step 2 (Approximating $r_b$).** Observe that the estimator (36) is an average of $N_b$ random variables bounded by the interval $[0, U]$. Let $E_{\mathcal{B}(x)}$ be the event that $(\mathcal{B}(X_1), \ldots, \mathcal{B}(X_n)) = (\mathcal{B}(x_1), \ldots, \mathcal{B}(x_n))$. On the event $E_{\mathcal{B}(x)}$, within each region $\mathcal{X}_b$, the number of point from the calibration set is known and the $Y_i$'s in each bin represent independent Bernoulli random variables that share the same mean $\mathbb{E}\left[ w(X) \mid X \in \mathcal{X}_b \right]$. Consider any fixed region $\mathcal{X}_b$, $b \in [B]$. By Hoeffding's inequality, it holds that

$$\mathbb{P}\left( |r_b - \widehat{r}_b| > \sqrt{\frac{U^2 \ln(6B/\alpha)}{2N_b}} \; \middle| \; E_{\mathcal{B}(x)} \right) \leqslant \alpha/(3B).$$

Applying union bound across all regions of the sample-space partition, we get that:

$$\mathbb{P}\left( \exists b \in [B] : \; |r_b - \widehat{r}_b| > \sqrt{\frac{U^2 \ln(6B/\alpha)}{2N_b}} \; \middle| \; E_{\mathcal{B}(x)} \right) \leqslant \alpha/3.$$

Because this is true for any $E_{\mathcal{B}(x)}$, we can marginalize to obtain that with probability at least $1 - \alpha/3$,

$$\forall b \in [B], \; |r_b - \widehat{r}_b| \leqslant \sqrt{\frac{U^2 \ln(6B/\alpha)}{2N_b}}. \qquad (37)$$

**Step 3 (Going from $r_b$ to $m_b$).** Define $r^\star = \min_{b\in[B]} \mathbb{E}\left[w(X) \mid X \in \mathcal{X}_b\right]$. Suppose $\forall b \in [B]$, $|r_b - \widehat{r}_b| \leqslant \varepsilon$ and $\varepsilon < r^\star/2$. Then, we have with probability at least $1 - \alpha/3$:

$$|m_b - \widehat{m}_b| = \left|\frac{1}{r_b} - \frac{1}{\widehat{r}_b}\right| = \left|\frac{r_b - \widehat{r}_b}{r_b \cdot \widehat{r}_b}\right| \leqslant \frac{\varepsilon}{r_b^2 |1 - \varepsilon/r_b|} \leqslant \frac{2\varepsilon}{r_b^2} = 2m_b^2 \varepsilon, \quad \forall b \in [B]. \qquad (38)$$

We now set $\varepsilon = \sqrt{\frac{U^2 \ln(6B/\alpha)}{2N_b}}$ as specified in equation (37) and verify that $\varepsilon < r^\star/2$.

- First, from step 1, with probability at least $1 - \alpha/3$, $N_{b^\star} = \Omega(n/B)$ and thus $N_b = \Omega(n/B)$ for every $b \in [B]$.

- By the condition in the theorem statement, for every $b \in [B]$,

$$\varepsilon = \sqrt{\frac{U^2 \ln(6B/\alpha)}{2N_b}} = O\left(\sqrt{\frac{U^2 B \ln(6B/\alpha)}{n}}\right) = O\left(\sqrt{\frac{U^2 B \ln(6B/\alpha)}{\left(\frac{U^2 B \ln(6B/\alpha)}{L^2}\right)}}\right) = O\left(L\right).$$

  Finally recall that $L \leqslant r^\star$. Thus we can pick $c$ in the theorem statement to be large enough such that $\varepsilon < L/2 \leqslant r^\star/2$.

Thus for $\varepsilon = \sqrt{\frac{U^2 \ln(6B/\alpha)}{2N_b}}$, by a union bound over the event in (37) and step 1, the conditions for (38) are satisfied with probability at least $1 - 2\alpha/3$. Hence we have for some large enough constant $c > 0$,

$$|m_b - \widehat{m}_b| \leqslant cm_b^2 \cdot \sqrt{\frac{U^2 B \ln(6B/\alpha)}{2n}} \leqslant c \cdot \frac{U}{L^2} \sqrt{\frac{B \ln(6B/\alpha)}{2n}}.$$

The final inequality holds by observing that $m_b \leqslant 1/L$ which follows from relationship (34) and the assumption that $\inf_x w(x) \geqslant L$.

**Step 4 (Computing the final deviation inequality for $\breve{\pi}_b^{(w)}$).** Recall the definitions of the two estimators:

$$\widehat{\pi}_b^{(w)} := \frac{1}{N_b} \sum_{i:\mathcal{B}(X_i)=b} m_b w(X_i) Y_i,$$

and

$$\breve{\pi}_b^{(w)} := \frac{1}{N_b} \sum_{i:\mathcal{B}(X_i)=b} \widehat{m}_b w(X_i) Y_i,$$

which differ by replacing $m_b$ by its estimator $\widehat{m}_b$ defined in (35). By triangle inequality,

$$\left|\breve{\pi}_b - \mathbb{E}\left[Y \mid X \in \mathcal{X}_b\right]\right| \leqslant \left|\breve{\pi}_b^{(w)} - \widehat{\pi}_b^{(w)}\right| + \left|\widehat{\pi}_b^{(w)} - \mathbb{E}\left[Y \mid X \in \mathcal{X}_b\right]\right|.$$

Theorem 8 bounds the term $\left|\widehat{\pi}_b^{(w)} - \mathbb{E}\left[Y \mid X \in \mathcal{X}_b\right]\right|$ with high probability. In the proof of Theorem 8, we can replace the empirical Bernstein's inequality by Hoeffding's inequality to obtain with probability at least $1 - \alpha/3$,

$$\left|\widehat{\pi}_b^{(w)} - \mathbb{E}\left[Y \mid X \in \mathcal{X}_b\right]\right| \leqslant \sqrt{\frac{U^2 \ln(6B/\alpha)}{2N_b}} \leqslant \left(\frac{U}{L}\right)^2 \sqrt{\frac{\ln(6B/\alpha)}{2N_b}},$$

simultaneously for all $b \in [B]$ (the last inequality follows since $L \leqslant 1 \leqslant U$). To bound $\left|\widehat{\pi}_b^{(w)} - \breve{\pi}_b^{(w)}\right|$, first note that:

$$\left|\widehat{\pi}_b^{(w)} - \breve{\pi}_b^{(w)}\right| = \left|\frac{1}{N_b} \sum_{i:\mathcal{B}(X_i)=b} (\widehat{m}_b - m_b) w(X_i) Y_i\right|$$

$$\leqslant U \cdot \left|\frac{1}{N_b} \sum_{i:\mathcal{B}(X_i)=b} (\widehat{m}_b - m_b)\right|$$

$$= U \cdot |\widehat{m}_b - m_b|.$$

Then we use the results from steps 1 and 3 to conclude that with probability at least $1 - 2\alpha/3$,

$$\left| \breve{\pi}_b^{(w)} - \widehat{\pi}_b^{(w)} \right| \leqslant c \cdot \left( \frac{U}{L} \right)^2 \sqrt{\frac{B \ln(6B/\alpha)}{2n}}, \quad \text{and} \quad N_b \geqslant n/B - \sqrt{\frac{n \ln(6B/\alpha)}{2}}.$$

simultaneously for all $b \in [B]$. Thus by union bound, we get that it holds with probability at least $1 - \alpha$,

$$|\breve{\pi}_b - \mathbb{E}\left[Y \mid X \in \mathcal{X}_b\right]| \leqslant c \cdot \left( \frac{U}{L} \right)^2 \sqrt{\frac{B \ln(6B/\alpha)}{2n}},$$

simultaneously for all $b \in [B]$ and large enough absolute constant $c > 0$. This concludes the proof. $\square$

### D.2 Proof of Theorem 8

Consider the event $E_{\mathcal{B}(x)}$ defined as $(\mathcal{B}(X_1), \ldots, \mathcal{B}(X_n)) = (\mathcal{B}(x_1), \ldots, \mathcal{B}(x_n))$. Conditioned on $E_{\mathcal{B}(x)}$, since $\sup_x w(x) \leqslant U$, we get that $\widehat{\pi}_b^{(w)}$ is an average of independent non-negative random variables $m_b w(X_i) Y_i$ that are bounded by $m_b U$ and share the same mean $m_b \mathbb{E}_P\left[w(X)Y \mid X \in \mathcal{X}_b\right] = \mathbb{E}_{\widetilde{P}}\left[Y \mid X \in \mathcal{X}_b\right]$ (by Proposition 3).Using Theorem 10 for a fixed $b \in [B]$, we obtain:

$$\mathbb{P}\left( \left| \widehat{\pi}_b^{(w)} - \mathbb{E}_{\widetilde{P}}\left[Y \mid X \in \mathcal{X}_b\right] \right| > \sqrt{\frac{2\widehat{V}_b \ln(3B/\alpha)}{N_b}} + \frac{3m_b U \ln(3B/\alpha)}{N_b} \,\Big|\, E_{\mathcal{B}(x)} \right) \leqslant \alpha/B.$$

Applying a union bound over all $b \in [B]$, we get:

$$\mathbb{P}\left( \forall b \in [B] : \left| \widehat{\pi}_b^{(w)} - \mathbb{E}_{\widetilde{P}}\left[Y \mid X \in \mathcal{X}_b\right] \right| \leqslant \sqrt{\frac{2\widehat{V}_b \ln(3B/\alpha)}{N_b}} + \frac{3m_b U \ln(3B/\alpha)}{N_b} \,\Big|\, E_{\mathcal{B}(x)} \right) \geqslant 1 - \alpha.$$

Because this is true for any $E_{\mathcal{B}(x)}$, we can marginalize to obtain the assertion of the theorem in unconditional form. $\square$

### D.3 Proof of Proposition 2

Fix any $\alpha \in (0,1)$. For any $k \in \mathbb{N}$ observe that by triangle inequality,

$$\left| \breve{\pi}_b^{(\widehat{w}_k)} - \mathbb{E}_{\widetilde{P}}\left[Y \mid X \in \mathcal{X}_b\right] \right| \leqslant \left| \breve{\pi}_b^{(w)} - \mathbb{E}_{\widetilde{P}}\left[Y \mid X \in \mathcal{X}_b\right] \right| + \left| \breve{\pi}_b^{(w)} - \breve{\pi}_b^{(\widehat{w}_k)} \right|.$$

Consider any $\varepsilon > 0$. Note that by Theorem 7, there exists sufficiently large $n$ such that the first term is larger than $\varepsilon/2$ with probability at most $\alpha/2$ simultaneously for all $b \in [B]$. Hence, it suffices to show that there exists a large enough $k$ such that the probability of the second term exceeding $\varepsilon/2$ is at most $\alpha/2$ simultaneously for all $b \in [B]$. While analyzing the second term, we treat $n$ as a constant while leveraging the consistency of $\widehat{w}_k$ as $k \to \infty$. For simplicity, denote $\Delta_k = \sup_x |w(x) - \widehat{w}_k(x)|$. Then for any $b \in [B]$:

$$\begin{aligned}
\left| \breve{\pi}_b^{(w)} - \breve{\pi}_b^{(\widehat{w}_k)} \right| &= \left| \frac{\sum_{i:\mathcal{B}(X_i)=b} w(X_i)Y_i}{\sum_{i:\mathcal{B}(X_i)=b} w(X_i)} - \frac{\sum_{i:\mathcal{B}(X_i)=b} \widehat{w}_k(X_i)Y_i}{\sum_{i:\mathcal{B}(X_i)=b} \widehat{w}_k(X_i)} \right| \\
&\stackrel{(1)}{\leqslant} \left| \frac{\sum_{i:\mathcal{B}(X_i)=b} w(X_i)Y_i}{\sum_{i:\mathcal{B}(X_i)=b} w(X_i)} - \frac{\sum_{i:\mathcal{B}(X_i)=b} \widehat{w}_k(X_i)Y_i}{\sum_{i:\mathcal{B}(X_i)=b} w(X_i)} \right| \\
&\quad + \left| \frac{\sum_{i:\mathcal{B}(X_i)=b} \widehat{w}_k(X_i)Y_i}{\sum_{i:\mathcal{B}(X_i)=b} w(X_i)} - \frac{\sum_{i:\mathcal{B}(X_i)=b} \widehat{w}_k(X_i)Y_i}{\sum_{i:\mathcal{B}(X_i)=b} \widehat{w}_k(X_i)} \right| \\
&\stackrel{(2)}{\leqslant} n \cdot \Delta_k \cdot \left| \frac{1}{\sum_{i:\mathcal{B}(X_i)=b} w(X_i)} \right|
\end{aligned}$$

$$+ \left| \frac{1}{\sum_{i:\mathcal{B}(X_i)=b} w(X_i)} - \frac{1}{\sum_{i:\mathcal{B}(X_i)=b} \widehat{w}_k(X_i)} \right| \left| \sum_{i:\mathcal{B}(X_i)=b} \widehat{w}_k(X_i)Y_i \right|$$

$$\overset{(3)}{\leqslant} \frac{n}{L} \cdot \Delta_k + \left( \frac{n \cdot \Delta_k}{(L - \Delta_k)L} \right) \cdot ((U + \Delta_k) \cdot n),$$

where (1) is due to the triangle inequality, (2) is due to the facts that the number of points in any bin is at most $n$ and that absolute difference between $\widehat{w}$ and $w$ is at most $\Delta_k$, (3) combines the aforementioned reasons in (2) and the assumptions: $L \leqslant \inf_x w(x) \leqslant \sup_x w(x) \leqslant U$. Since $\Delta_k \overset{P}{\to} 0$, clearly there exists a large enough $k$ such that:

$$\mathbb{P}\left( \left| \breve{\pi}_b^{(w)} - \breve{\pi}_b^{(\widehat{w}_k)} \right| \geqslant \varepsilon/2 \right) \leqslant \alpha/2.$$

Thus we conclude that $\breve{\pi}_b^{(\widehat{w}_k)}$ is asymptotically calibrated at level $\alpha$. $\qquad\square$

### D.4  Preliminary simulations

This section is structured as follows. We first describe the overall procedure for calibration under covariate shift. The finite-sample calibration guarantee of Theorem 7 holds for oracle $w$ whereas in our experiments we will estimate $w$; to assess the loss in calibration due to this approximation, we introduce some standard techniques used in literature. The preliminary experiments are performed with simulated data which are described after this. Finally, we propose a modified estimator $\widetilde{\pi}_b^{(\widehat{w})}$ of $\mathbb{E}_{\widetilde{P}}[Y \mid X \in \mathcal{X}_b]$ which appears natural but has poor performance in practice.

**Procedure.**   We describe how to construct approximately calibrated predictions practically. This involves approximating the importance weights $w$ and the relatives mass terms $\{m_b\}_{b\in[B]}$. The summarized calibration procedure consists of the following steps:

1. Split the calibration set into two parts and use the first to perform *uniform mass* binning
2. Given unlabeled examples from both source and target domain, estimate $\widehat{w}$. The unconstrained Least-Squares Importance Fitting (uLSIF) procedure [17] is used for this.
3. Compute for every $b \in [B]$, the estimator as per (17), replacing $w$ with $\widehat{w}$:

$$\breve{\pi}_b^{(\widehat{w})} := \frac{\sum_{i:\mathcal{B}(X_i)=b} \widehat{w}(X_i)Y_i}{\sum_{i:\mathcal{B}(X_i)=b} \widehat{w}(X_i)}. \tag{39}$$

4. On a new test point from the target distribution, output the calibrated estimate $\breve{\pi}_{\mathcal{B}(X_{n+1})}^{(\widehat{w})}$.

**Assessment through reliability diagrams and ECE.**   Given a test set (from the target distribution) of size $m$: $\{(X_i', Y_i')\}_{i\in[m]}$ and a function $g : \mathcal{X} \to [0, 1]$ that outputs approximately calibrated probabilities, we consider the reliability diagram to estimate its calibration properties. A reliability diagram is constructed using splitting the unit interval $[0, 1]$ into non-overlapping intervals $\{I_b\}_{b\in[B']}$ for some $B'$ as

$$I_i = \left[ \frac{i-1}{B'}, \frac{i}{B'} \right), \ i = 1, \ldots, B'-1 \text{ and } I_{B'} = \left[ \frac{B'-1}{B'}, 1 \right].$$

Let $\mathcal{B}' : [0, 1] \to [B']$ denote the binning function that corresponds to this binning. We then compute the following quantities for each bin $b \in [B']$:

$$\text{FP}(I_b) = \frac{\sum_{i:\mathcal{B}'(X_i')=b} Y_i'}{|\{i : \mathcal{B}'(X_i') = b\}|} \qquad \text{(fraction of positives in a bin)},$$

$$\text{MP}(I_b) = \frac{\sum_{i:\mathcal{B}'(X_i')=b} g(X_i')}{|\{i : \mathcal{B}'(X_i') = b\}|} \qquad \text{(mean predicted probability in a bin)}.$$

If $g$ is perfectly calibrated, the reliability diagram is diagonal. Define the proportion of points that fall into various bins as:

$$\widehat{p}_b = \frac{|\{i : \mathcal{B}'(X_i') = b\}|}{m}, \quad b \in [B'].$$

Figure 2: In Figure 2a uncalibrated Random Forest (ECE $\approx 0.023$) is compared with calibration that does not take the covariate shift into account (ECE $\approx 0.047$). In Figure 2b uncalibrated Random Forest is compared with calibration that takes the covariate shift into account (ECE $\approx 0.017$).

Then ECE (or $\ell_1$-ECE) is defined as:

$$\text{ECE}(g) = \sum_{b \in [B']} \widehat{p}_b \cdot |\text{MP}(I_b) - \text{FP}(I_b)| \, .$$

ECE can also be defined in the $\ell_p$ sense and for multiclass problems but we limit our attention to the $\ell_1$-ECE for binary problems.

**Simulations with synthetic data.** We illustrate the performance of our proposed estimator (17) using the following simulated example, for which we can explicitly control the covariate shift. Consider the following data generation pipeline: for the source domain each component of the feature vector is drawn from $\text{Beta}(\alpha, \beta)$ where $\alpha = \beta = 1$, which corresponds to uniform draws from the unit cube. For the target distribution each component can be drawn independently from $\text{Beta}(\alpha', \beta')$. If the dimension is $d$, the true likelihood ratio is given as

$$w(x) = \frac{d\widetilde{P}_X(x)}{dP_X(x)} = \frac{B^d(\alpha; \beta)}{B^d(\alpha'; \beta')} \prod_{i=1}^{d} \frac{(x_{(i)})^{\alpha'-1}(1 - x_{(i)})^{\beta'-1}}{(x_{(i)})^{\alpha-1}(1 - x_{(i)})^{\beta-1}},$$

where $x_{(i)}$ are the coordinates of feature vector $x$. We set $d = 3$ and $\alpha' = 2, \beta' = 1$ so that $w(x) = 8 \cdot x_{(1)} x_{(2)} x_{(3)}$. The labels for both source and target distributions are assigned according to:

$$\mathbb{P}(Y = 1 \mid X = x) = \frac{1}{2} \left( 1 + \sin \left( \omega \left( x_{(1)}^2 + x_{(2)}^2 + x_{(3)}^2 \right) \right) \right),$$

for $\omega = 20$. As the underlying classifier we use a Random Forest with 100 trees (from `sklearn`). 14700 data points were used to train the underlying Random Forest classifier, 2000 data points from both source and target were used for the estimation of importance weights. The parameters $\sigma$ and $\lambda$ for uLSIF were tuned by leave-one-out cross-validation: we considered 25 equally spaced values on a log-scale in range $(10^{-2}, 10^2)$ for $\sigma$ and 100 equally spaced values on a log-scale in range $(10^{-3}, 10^3)$ for $\lambda$. Uniform mass binning was performed with 10 bins and 1940 data points from the source domain were used to estimate the quantiles. 7840 source data points were used for the calibration and finally, 28000 data points from the target domain were used for evaluation purposes. We note that this simulation is a 'proof-of-concept'; the sample sizes we used are not necessarily optimal can presumably be improved.

We compare the unweighted estimator (12) which corresponds to weighing points in each bin equally as we would do if there was no covariate shift, and the estimator (17) that uses an estimate of $w$ to account for covariate shift. The reliability diagrams are presented in Figure 2, with the ECE reported in the caption. For the ECE estimation and reliability diagrams, we used $B' = 10$.

Figure 3: Calibration of Random Forest with $m_b$ estimated as per equation (35) (ECE $\approx 0.05$).

**Alternative estimator for** $m_b$. Estimator (35) is one way of estimating $m_b$ using the $w$ values, that leads to (17). However, there exists another natural estimator which we propose and show some preliminary empirical results for. Suppose we have access to additional unlabeled data from the source and target domains ($\{X_i^s\}_{i \in [n_s]}$, and $\{X_i^t\}_{i \in [n_t]}$ respectively). From the definition of $m_b = \mathbb{P}_{P_X}(X \in \mathcal{X}_b)/\mathbb{P}_{\tilde{P}_X}(X \in \mathcal{X}_b)$, a natural estimator is,

$$\widehat{m}_b = \frac{\frac{1}{n_s}|\{i \in [n_s] : \mathcal{B}(X_i^s) = b\}|}{\frac{1}{n_t}|\{i \in [n_t] : \mathcal{B}(X_i^t) = b\}|}, \quad b \in [B]. \tag{40}$$

In this case, the estimator (33) reduces to:

$$\widetilde{\pi}_b^{(\widehat{w})} = \frac{\widehat{m}_b}{N_b} \sum_{i:\mathcal{B}(X_i)=b} \widehat{w}(X_i)Y_i.$$

We show experimental results with this estimation procedure. We used 8500 data points from the source domain and 8000 points from the target domain to compute (40). The reliability diagram and ECE with this estimator is reported in Figure 3. On our simulated dataset, we observe that the estimators $\widetilde{\pi}_b^{(\widehat{w})}$ perform significantly worse than the estimators $\widecheck{\pi}_b^{(\widehat{w})}$. While this is only a single experimental setup, we outline some drawbacks of this estimation method that may lead to poor performance in general.

1. $\widetilde{\pi}_b^{(\widehat{w})}$ requires access to additional unlabeled data from the source and target domains without leading to increase in performance.

2. The denominator of $\widehat{m}_b$ could be badly behaved if the number of points from the target domain in bin $b$ are small. We could perform uniform-mass binning on the target domain to avoid this, but in this case $N_b$ may be small which would lead to the estimator $\widetilde{\pi}_b^{(\widehat{w})}$ performing poorly.

Our overall recommendation through these preliminary experiments is to use the estimator $\widehat{\pi}_b^{(\widehat{w})}$ as proposed in Section 4.4 instead of $\widetilde{\pi}_b^{(\widehat{w})}$.

# E  Venn prediction

Venn prediction [24, 45–47] is a calibration framework that provides distribution-free guarantees, which are different from the ones in Definitions 1 and 2. For a multiclass problem with $L$ labels, Venn prediction produces $L$ predictions, one of which is guaranteed to be perfectly calibrated

(although it is impossible to know which one). These are called multiprobabilistic predictors, formally defined as a collection of predictions $(f_1, f_2, \ldots f_L)$ where each $f_i \in \{\mathcal{X} \rightarrow \Delta_{L-1}\}$ (here $\Delta_{L-1}$ is the boundary of the $\ell_1$ ball in the non-negative orthant of $\mathbb{R}^L$, corresponding to all possible distributions over $\{1, 2, \ldots, L\}$). Vovk and Petej [45] defined two calibration guarantees for multiprobabilistic predictors, the first being oracle calibration.

**Definition 4** (Oracle calibration). $(f_1, f_2, \ldots f_L)$ is oracle calibrated if there exists an oracle selector $S$ such that $f_S$ is perfectly calibrated.

Venn predictors satisfy oracle calibration [45, Theorem 1] with $S = Y$. In the binary case, this means that when $Y = 1$, $f_1(X)$ is perfectly calibrated but we do not have any guarantee on $f_0(X)$; on the other hand if $Y = 0$, $f_0(X)$ is perfectly calibrated but we know nothing about $f_1(X)$. Since $Y$ is unknown, oracle calibration seems to us to primarily serve as theoretical guidance, but does not give a clear prescription on what to output and what theoretical guarantee that output satisfies. In practice, it seems reasonable to suspect that if $f_0(X)$ and $f_1(X)$ are close, then their average should be approximately calibrated in the sense of Definition 1, but to the best of our knowledge, such results have not been shown formally (other aggregate functions apart from average are also suggested (without formal guarantees) by Vovk and Petej [45, Section 4]). For instance, it may be tempting to think that oracle calibration of a multiprobabilistic predictor leads to approximate calibration in the following way. Consider the prediction function

$$f(X) = \frac{\min f_i(X) + \max f_i(X)}{2},$$

and the radius of the interval $[\min f_i(X), \max f_i(X)]$:

$$\varepsilon(X) = \frac{\max f_i(X) - \min f_i(X)}{2}.$$

Since Venn predictors satisfy oracle calibration, one might conjecture that $f$ is $(\varepsilon, \alpha)$ approximately calibration (per Definition 1) for the given function $\varepsilon$ and for any $\alpha \in (0, 1)$. We examined this claim but were unable to prove such a guarantee formally. In fact, it seems that no general calibration guarantee should be possible with the size of the calibration interval being $O(\varepsilon(X))$; we evidence this through the following construction.

Consider a setup, with no covariates and only label values $Y$, and a single bin that contains all points (in the Venn prediction language: a taxonomy under which all points are equivalent). For a test-point $Y_{n+1}$ and any predictor $f$, note that $\mathbb{E}[Y_{n+1} \mid f]$ is simply equal to $\mathbb{E}[Y_{n+1}]$ since any information used to construct $f$ is independent of $Y_{n+1}$. To ensure calibration, we may look for a guarantee of the following form for some $\delta$:

$$|\mathbb{E}[Y_{n+1} \mid f] - f| = |\mathbb{E}[Y_{n+1}] - f| \leqslant \delta.$$

In essence, $f$ is an estimator for the parameter $\mathbb{E}[Y]$ with a corresponding deviation bound of $\delta$. Without distributional assumptions, we only expect to estimate such a parameter with error at best $\delta = O(1/\sqrt{n})$ for a fixed constant probability of failure. On the other hand, the Venn prediction interval $[\min f_i, \max f_i]$ often has radius $O(1/n)$. Thus for valid approximate calibration, we would need to provide a larger interval than $[\min f_i, \max f_i]$, even though one of the $f_i$'s is perfectly calibrated. Given this example, our conjecture is that it might be possible to show that there always exists an $f_i(X)$ that is $(n^{-0.5}\text{polylog}(1/\alpha)), \alpha)$ calibrated. Without knowing which $f_i(X)$ to pick, perhaps one can show that an aggregate point in the interval $[\min f_i, \max f_i]$ is $((\max f_i - \min f_i) + n^{-0.5}\text{polylog}(1/\alpha), \alpha)$ approximately calibrated. In Section 4, we showed such a result for histogram binning (which can be interpreted as a Venn predictor). It would be interesting to study if such results can be shown for general Venn predictors.

Another guarantee for multiprobabilistic predictors is calibration in the large.

**Definition 5** (Calibration in the large). $(f_1, f_2, \ldots f_L)$ is calibrated in the large if the following is satisfied: $\mathbb{E}[Y] \in [\mathbb{E}\min f_i(X), \mathbb{E}\max f_i(X)]$.

Vovk and Petej [45, Theorem 2] show that Venn predictors satisfy calibration in the large. Due to the expectation signs and the coverage of the marginal probability $\mathbb{E}[Y]$, calibration in the large does not lead to a clear interpretable guarantee for uncertainty quantification, but rather a minimum requirement that serves as a guiding principle.

# F  Auxiliary results

## F.1  Concentration inequalities

**Theorem 9** (Howard et al. [15], Theorem 4). *Suppose $Z_t \in [a, b]$ a.s. for all $t$. Let $(\widehat{Z}_t)$ be any $[a, b]$-valued predictable sequence, and let $\mathfrak{u}$ be any sub-exponential uniform boundary with crossing probability $\alpha$ for scale $c = b - a$. Then:*

$$\mathbb{P}\left( \forall t \geqslant 1 : \left| \overline{Z}_t - \mu_t \right| < \frac{\mathfrak{u}\left( \sum_{i=1}^{t} \left( Z_i - \widehat{Z}_i \right)^2 \right)}{t} \right) \geqslant 1 - 2\alpha.$$

**Theorem 10** (Partial statement of Audibert et al. [2], Theorem 1). *Let $X_1, \ldots, X_n$ be i.i.d. random variables bounded in $[0, s]$, for some $s > 0$. Let $\mu = \mathbb{E}[X_1]$ be their common expected value. Consider the empirical mean $\overline{X}_n$ and variance $V_n$ defined respectively by*

$$\overline{X}_n = \frac{\sum_{i=1}^{n} X_i}{n}, \quad \text{and} \quad V_n = \frac{\sum_{i=1}^{n} (X_i - \overline{X}_n)^2}{n}.$$

*Then for any $\delta \in (0, 1)$, with probability at least $1 - \delta$,*

$$\left| \overline{X}_n - \mu \right| \leqslant \sqrt{\frac{2 V_n \log(3/\delta)}{n}} + \frac{3s \log(3/\delta)}{n}.$$

## F.2  Uniform-mass binning

Kumar et al. [21] defined well-balanced binning and showed that uniform mass-binning is well-balanced.

**Definition 6** (Well-balanced binning). A binning scheme $\mathcal{B}$ of size $B$ is $\beta$-well-balanced ($\beta \geqslant 1$) for some classifier $g$ if

$$\frac{1}{\beta B} \leqslant \mathbb{P}\left( g(X) \in I_b \right) \leqslant \frac{\beta}{B},$$

simultaneously for all $b \in [B]$.

To perform uniform-mass binning labeled examples are required at the stage of training the base classifier $g(\cdot)$. We denote this data as $\mathcal{D}_{\text{cal}}^1$. Procedures based on uniform-mass binning are well-balanced if $\left| \mathcal{D}_{\text{cal}}^1 \right|$ is sufficiently large.

**Lemma 11** (Kumar et al. [21], Lemma 4.3). *For a universal constant $c > 0$, if $\left| \mathcal{D}_{cal}^1 \right| \geqslant cB \ln(B/\alpha)$, then with probability at least $1 - \alpha$, the uniform mass binning scheme $\mathcal{B}$ is 2-well-balanced.*

The calibration guarantees in Section 4 depend on the minimum number of training points $N_{b^\star}$ in any bin. Uniform mass-binning guarantees that $N_{b^\star} = \Omega(n/B)$. This is used in the proof of Theorem 5.

[Supplementary Material 2]

# Appendix

The Appendix contains proofs of results in the main paper ordered as they appear. Auxiliary results needed for some of the proofs are stated in Appendix F.

## A  Proof of Proposition 1

The 'if' part of the theorem is due to Vaicenavicius et al. [44, Proposition 1]; we reproduce it for completeness. Let $\sigma(g), \sigma(f)$ be the sub $\sigma$-algebras generated by $g$ and $f$ respectively. By definition of $f$, we know that $f$ is $\sigma(g)$-measurable and, hence, $\sigma(f) \subseteq \sigma(g)$. We now have:

$$
\begin{aligned}
\mathbb{E}\left[Y \mid f(X)\right] &= \mathbb{E}\left[\mathbb{E}\left[Y \mid g(X)\right] \mid f(X)\right] && \text{(by tower rule since } \sigma(f) \subseteq \sigma(g)) \\
&= \mathbb{E}\left[f(X) \mid f(X)\right] && \text{(by property (5))} \\
&= f(X).
\end{aligned}
$$

The 'only if' part can be verified for $g = f$. Since $f$ is perfectly calibrated,

$$
\mathbb{E}\left[Y \mid f(X) = f(x)\right] = f(x),
$$

almost surely $P_X$.

$\square$

## B  Proofs of results in Section 3

### B.1  Proof of Theorem 1

Assume that one is given a predictor $f$ that is $(\varepsilon, \alpha)$-approximately calibrated. Then the assertion follows from the definition of $(\varepsilon, \alpha)$-approximate calibration since:

$$
\left|\mathbb{E}\left[Y \mid f(X)\right] - f(X)\right| \leqslant \varepsilon(f(X)) \implies \mathbb{E}\left[Y \mid f(X)\right] \in C(f(X)).
$$

Now we show the proof in the other direction. If $m_C$ was injective, $\mathbb{E}\left[Y \mid m_C(f(X))\right] = \mathbb{E}\left[Y \mid f(X)\right]$ and thus if $\mathbb{E}\left[Y \mid f(X)\right] \in C(f(X))$ (which happens with probability at least $1 - \alpha$), we would have $\mathbb{E}\left[Y \mid m_C(f(X))\right] \in C(f(X))$ and so

$$
\left|\mathbb{E}\left[Y \mid m_C(f(X))\right] - m_C(f(X))\right| \leqslant \sup_{z \in \text{Range}(f)} \{|C(z)|/2\} = \varepsilon.
$$

This serves as an intuition for the proof in the general case, when $m_C$ need not be injective. Note that,

$$
\begin{aligned}
\left|\mathbb{E}\left[Y \mid m_C(f(X))\right] - m_C(f(X))\right| &= \left|\mathbb{E}\left[Y \mid m_C(f(X))\right] - \mathbb{E}\left[m_C(f(X)) \mid m_C(f(X))\right]\right| \\
&\stackrel{(1)}{=} \left|\mathbb{E}\left[\mathbb{E}\left[Y \mid f(X)\right] \mid m_C(f(X))\right] - \mathbb{E}\left[m_C(f(X)) \mid m_C(f(X))\right]\right| \\
&\stackrel{(2)}{=} \left|\mathbb{E}\left[\mathbb{E}\left[Y \mid f(X)\right] - m_C(f(X)) \mid m_C(f(X))\right]\right| \\
&\stackrel{(3)}{\leqslant} \mathbb{E}\left[\left|\mathbb{E}\left[Y \mid f(X)\right] - m_C(f(X))\right| \mid m_C(f(X))\right], && (20)
\end{aligned}
$$

where we use the tower rule in (1) (since $m_C$ is a function of $f$), linearity of expectation in (2) and Jensen's inequality in (3). To be clear, the outermost expectation above is over $f(X)$ (conditioned on $m_C(f(X))$). Consider the event

$$
A : \mathbb{E}\left[Y \mid f(X)\right] \in C(f(X)).
$$

On $A$, by definition we have:

$$
\left|\mathbb{E}\left[Y \mid f(X)\right] - m_C(f(X))\right| = \frac{u_C(f(X)) - l_C(f(X))}{2} \leqslant \sup_{z \in \text{Range}(f)} \left(\frac{|C(z)|}{2}\right) = \varepsilon.
$$

By monotonicity property of conditional expectation, we also have that conditioned on $A$,

$$
\mathbb{E}\left[\left|\mathbb{E}\left[Y \mid f(X)\right] - m_C(f(X))\right| \mid m_C(f(X))\right] \leqslant \mathbb{E}\left[\varepsilon \mid m_C(f(X))\right] = \varepsilon,
$$

with probability 1. Thus by the relationship proved in the series of equations ending in (20), we have that conditioned on $A$, with probability 1,

$$|\mathbb{E}\left[Y \mid m_C(f(X))\right] - m_C(f(X))| \leqslant \varepsilon.$$

Since we are given that $C$ is a $(1-\alpha)$-CI with respect to $f$, $\mathbb{P}(A) \geqslant 1-\alpha$. For any event $B$, it holds that $\mathbb{P}(B) \geqslant \mathbb{P}(B|A)\mathbb{P}(A)$. Setting

$$B : |\mathbb{E}\left[Y \mid m_C(f(X))\right] - m_C(f(X))| \leqslant \varepsilon,$$

we obtain:

$$\mathbb{P}\left(|\mathbb{E}\left[Y \mid m_C(f(X))\right] - m_C(f(X))| \leqslant \varepsilon\right) \geqslant 1-\alpha.$$

Thus, we conclude that $m_C(f(\cdot))$ is $(\varepsilon, \alpha)$-approximately calibrated. $\qquad\square$

## B.2   Proof of Corollary 1

Let $\{f_n\}_{n\in\mathbb{N}}$ be asymptotically calibrated sequence with the corresponding sequence of functions $\{\varepsilon_n\}_{n\in\mathbb{N}}$ that satisfy $\varepsilon_n(f_n(X_{n+1})) = o_P(1)$. From Theorem 1, we can construct corresponding functions $C_n$ that are $(1-\alpha)$-CI with respect to $f_n$ and satisfy

$$|C_n(f_n(X_{n+1}))| = 2\varepsilon_n(f_n(X_{n+1})) = o_P(1).$$

This concludes the proof. $\qquad\square$

## B.3   Proof of Theorem 2

In the proof we write the test point as $(X_{n+1}, Y_{n+1})$. Suppose $\widehat{C}_n$ is a $(1-\alpha)$-CI with respect to $f$ for all distributions $P$. We show that $\widehat{C}_n$ covers the label $Y_{n+1}$ itself for distributions $P$ such that $P_{f(X)}$ is nonatomic (and thus $\text{disc}(\widehat{C}_n)$ would also cover the labels).

Let $P$ be any distribution such that $P_{f(X)}$ is nonatomic. Fix a set of $m \geqslant n+1$ samples from the distribution $P$ denoted as $\mathcal{T} = \{(A^{(j)}, B^{(j)})\}_{j\in[m]}$. Given $\mathcal{T}$, consider a distribution $Q$ corresponding to the following sampling procedure for $(X, Y) \sim Q$:

sample an index $j$ uniformly at random from $[m]$ and set $(X, Y) = (A^{(j)}, B^{(j)})$.

The distribution function for $Q$ is given by

$$m^{-1} \sum_{j=1}^{m} \delta_{(A^{(j)}, B^{(j)})}.$$

where $\delta_{(a,b)}$ denotes the points mass at $(a, b)$. Note that $Q$ is only defined conditional on $\mathcal{T}$. Observe the following facts about $Q$:

- $\text{supp}(Q) = \{(A^{(j)}, B^{(j)})\}_{j\in[m]}$.
- Consider any $(x, y) \in \text{supp}(Q)$. Let $(x, y) = (A^{(j)}, B^{(j)})$ for some $j \in [m]$. Then

$$\mathbb{E}_Q\left[Y \mid f(X) = f(x)\right] = \mathbb{E}_Q\left[Y \mid f(X) = f(A^{(j)})\right]$$

$$\overset{\xi_1}{=} \mathbb{E}_Q\left[Y \mid X = A^{(j)}\right]$$

$$\overset{\xi_2}{=} B^{(j)} = y.$$

  Above $\xi_1$ holds since $P_{f(X)}$ is nonatomic so that the $f(X^{(i)})$'s are unique almost surely. Note that $P_{f(X)}$ is nonatomic only if $P_X$ itself is nonatomic. Thus the $A^{(j)}$'s are unique almost surely, and $\xi_2$ follow. In other words, if $(X, Y) \sim Q$, then we have

$$Y = \mathbb{E}_Q\left[Y \mid f(X)\right]. \tag{21}$$

Suppose the data distribution was $Q$, that is $\{(X_i, Y_i)\}_{i \in [n+1]} \sim Q^{n+1}$. Define the event that the CI guarantee holds as

$$E_1 : \mathbb{E}\left[Y_{n+1} \mid f(X_{n+1})\right] \in \widehat{C}_n(f(X_{n+1})), \tag{22}$$

and the event that the PS guarantee holds as

$$E_2 : Y_{n+1} \in \widehat{C}_n(f(X_{n+1})). \tag{23}$$

Then due to (21), the events are exactly the same under $Q$:

$$E_1 \stackrel{Q}{\equiv} E_2. \tag{24}$$

In particular, this means

$$\mathbb{P}_{Q^{n+1}}(\mathbb{E}_Q\left[Y_{n+1} \mid f(X_{n+1})\right] \in \widehat{C}_n(f(X_{n+1}))) = \mathbb{P}_{Q^{n+1}}(Y_{n+1} \in \widehat{C}_n(f(X_{n+1}))). \tag{25}$$

If $\widehat{C}_n$ is a distribution-free CI, then $\mathbb{P}_{Q^{n+1}}(E_1) \geqslant 1 - \alpha$ and thus $\mathbb{P}_{Q^{n+1}}(E_2) \geqslant 1 - \alpha$. This shows that for $Q$, $\mathrm{disc}(\widehat{C}_n)$ is a $(1 - \alpha)$-PI. Note that $Q$ corresponds to sampling *with replacement* from a fixed set $\mathcal{T}$ where each element is drawn with respect to $P$. Although $Q \neq P$, we expect that as $m \to \infty$ (while $n$ is fixed), $Q$ and $P$ coincide. This would prove the result for general $P$. To formalize this intuition, we describe a distribution which is close to $Q$ but corresponds to sampling *without replacement* from $\mathcal{T}$ instead.

For this, now suppose that $\{(X_i, Y_i)\}_{i \in [n+1]} \sim R^{n+1}$ where $R^{n+1}$ corresponds to sampling without replacement from $\mathcal{T}$. Formally, to draw from $R^{n+1}$, we first draw a surjective mapping $\lambda : [n+1] \to [m]$ as

$$\lambda \sim \mathrm{Unif}\ (n\text{-sized ordered subsets of } [m]),$$

and set $(X_i, Y_i) = (A^{(\lambda(i))}, B^{(\lambda(i))})$ for $i \in [n + 1]$.

First we quantify precisely the intuition that as $m \to \infty$, $Q^{n+1}$ and $R^{n+1}$ are essentially identical. Consider the event $T := $ no index is repeated in $Q^{n+1}$. Let $\mathbb{P}(T) = \tau_m$ for some $m$ and note that $\lim_{m \to \infty} \tau_m = 1$. Now consider any probability event $E$ over $\{(X_i, Y_i)\}_{i \in [n+1]}$ (such as $E_1$ or $E_2$). We have

$$\mathbb{P}_{Q^{n+1}}(E) = \mathbb{P}_{Q^{n+1}}(E|T) \cdot \mathbb{P}(T) + \mathbb{P}_{Q^{n+1}}(E|T^c) \cdot \mathbb{P}(T^c)$$
$$\in \left[\mathbb{P}_{Q^{n+1}}(E|T) \cdot \mathbb{P}(T), \mathbb{P}_{Q^{n+1}}(E|T) \cdot \mathbb{P}(T) + \mathbb{P}(T^c)\right].$$

Now observe that $\mathbb{P}_{Q^{n+1}}(E|T) = \mathbb{P}_{R^{n+1}}(E)$ to conclude

$$\mathbb{P}_{Q^{n+1}}(E) \in \left[\mathbb{P}_{R^{n+1}}(E) \cdot \mathbb{P}(T), \mathbb{P}_{R^{n+1}}(E) \cdot \mathbb{P}(T) + \mathbb{P}(T^c)\right].$$

Since $m \geqslant n + 1$, $\mathbb{P}(T) \neq 0$ so we can invert the above and substitute $\tau_m = \mathbb{P}(T)$ to get

$$\mathbb{P}_{R^{n+1}}(E) \in \left[\tau_m^{-1}(\mathbb{P}_{Q^{n+1}}(E) - (1 - \tau_m)),\ \tau_m^{-1}\mathbb{P}_{Q^{n+1}}(E)\right]. \tag{26}$$

Consider $E = E_2$ defined in equation (23). We showed that $\mathbb{P}_{Q^{n+1}}(E_2) \geqslant 1 - \alpha$. Thus from (26),

$$\mathbb{P}_{R^{n+1}}(E_2) \geqslant \tau_m^{-1}(1 - \alpha - (1 - \tau_m)).$$

The above is with respect to $R^{n+1}$ which is conditional on a fixed draw $\mathcal{T}$. However since the right hand side is independent of $\mathcal{T}$, we can also include the randomness in $\mathcal{T}$ to say:

$$\mathbb{P}_{R^{n+1}, \mathcal{T}}(E_2) \geqslant \tau_m^{-1}(1 - \alpha - (1 - \tau_m)). \tag{27}$$

Observe that if we consider the marginal distribution over $R^{n+1}$ and $\mathcal{T}$ (that is we include the randomness in $\mathcal{T}$ as above), $\{(X_i, Y_i)\}_{i \in [n+1]} \stackrel{iid}{\sim} P$. This is not true if we do not marginalize over $\mathcal{T}$, in particular since the $(X_i, Y_i)$'s are not independent (due to sampling without replacement). Thus equation (27) can be restated as

$$\mathbb{P}_{P^{n+1}}(E_2) \geqslant \tau_m^{-1}(1 - \alpha - (1 - \tau_m)),$$

Since $m$ can be set to any number and $\lim_{m \to \infty} \tau_m = 1$, we can indeed conclude

$$\mathbb{P}_{P^{n+1}}(E_2) \geqslant 1 - \alpha.$$

Recall that $E_2$ is the event that $Y_{n+1} \in \widehat{C}_n(X_{n+1})$; equivalently $Y_{n+1} \in \mathrm{disc}\widehat{C}_n(X_{n+1})$. Thus $\mathrm{disc}(\widehat{C}_n)$ provides a $(1 - \alpha)$-PI for $P$ such that $P_{f(X)}$ is nonatomic. $\qquad\square$

## B.4 Proof of Corollary 2

Let $P$ be any distribution such that $P_{f(X)}$ is nonatomic. By Theorem 2, $\widehat{C}_n$ must provide both a prediction set and a confidence interval for $P$:

$$\mathbb{P}(\mathbb{E}\left[Y_{n+1} \mid f(X_{n+1})\right] \in \widehat{C}_n(f(X_{n+1}))) \geqslant 1 - \alpha,$$

and

$$\mathbb{P}(Y_{n+1} \in \widehat{C}_n(f(X_{n+1}))) \geqslant 1 - \alpha.$$

Thus by a union bound

$$\mathbb{P}_{P^{n+1}}(\{Y_{n+1}, \mathbb{E}\left[Y_{n+1} \mid f(X_{n+1})\right]\} \subseteq \widehat{C}_n(f(X_{n+1}))) \geqslant 1 - 2\alpha. \tag{28}$$

Now consider a distribution $P$ such that $P_{f(X)}$ is nonatomic and $\mathbb{P}(Y = 1 \mid X) = 0.5$ a.s. $P_X$ so that $\mathbb{E}\left[Y_{n+1} \mid f(X)\right] = 0.5$ a.s. $P_{f(X)}$. The inequality (28) is true for this $P$ as well. If

$$\{Y_{n+1}, \mathbb{E}\left[Y_{n+1} \mid f(X_{n+1})\right]\} \subseteq \widehat{C}_n(f(X_{n+1})),$$

then $|\widehat{C}_n(X_{n+1})| \geqslant |Y_{n+1} - \mathbb{E}\left[Y_{n+1} \mid f(X_{n+1})\right]| \geqslant 0.5$. Thus

$$\mathbb{P}_{P^{n+1}}(|\widehat{C}_n(f(X_{n+1}))| \geqslant 0.5) \geqslant 1 - 2\alpha.$$

Consequently we have

$$\mathbb{E}_{P^{n+1}}|\widehat{C}_n(f(X_{n+1}))| \geqslant 0.5(1 - 2\alpha)$$
$$= 0.5 - \alpha.$$

This concludes the proof. $\qquad\square$

## B.5 Proof of Theorem 3

Suppose that $\{f_n\}_{n\in\mathbb{N}}$ is asymptotically calibrated and satisfies

$$\limsup_{n\to\infty} \left|\mathcal{X}^{(f_n)}\right| > \aleph_0,$$

that is, for every $m \in \mathbb{N}$, there exists $n \geqslant m$ such that $\mathcal{X}^{(f_n)}$ is an uncountable set. We will show a contradiction using Corollary 2 for $f_n$ and a certain $C_n$ to be defined shortly.

First, we verify the condition of Corollary 2 for $f_n$ if $\mathcal{X}^{(f_n)}$ is uncountable: we construct a distribution $P$ such that $P_{(f_n(X))}$ is nonatomic. Let the range of $f_n$ acting on $\mathcal{X}$ be denoted as $f_n(\mathcal{X})$, and for $z \in f_n(\mathcal{X})$ let the level set at value $z$ be denoted as $\mathcal{X}_z^{(f_n)}$. Since the sets $\mathcal{X}^{(f_n)}$ are measurable, we can define $P(X)$ as follows:

$$P(f_n(X)) = \mathrm{Unif}(f_n(\mathcal{X})); \quad P(X \mid f_n(X)) = \mathrm{Unif}\left(\mathcal{X}_{f_n(X)}^{(f_n)}\right). \tag{29}$$

$P(X)$ along with any conditional probability function $P(Y \mid X)$ constitutes a valid probability distribution $P$. Further, from the construction, since $\mathcal{X}^{(f_n)}$ is uncountable, $P_{f_n(X)}$ is guaranteed to be nonatomic.

Next, since $\{f_n\}_{n\in\mathbb{N}}$ is asymptotically calibrated, by Corollary 1, one can construct a sequence of functions $\{C_n\}_{n\in\mathbb{N}}$ such that each $C_n$ is a $(1 - \alpha)$-CI with respect to $f_n$ for any distribution $Q$, and

$$|C_n(f_n(X_{n+1}))| = o_Q(1).$$

Thus there exists a constant $m$ such that for $n \geqslant m$ and any distribution $Q$,

$$\mathbb{E}_{Q^{n+1}}|C_n(f_n(X_{n+1}))| < 0.5 - \alpha. \tag{30}$$

However, since $\limsup_{n\to\infty}|\mathcal{X}^{(f_n)}| > \aleph_0$, there exists an $n \geqslant m$ such that $\mathcal{X}^{(f_n)}$ is uncountable. Hence the requirements of Corollary 2 are satisfied by $\widehat{C}_n$ and $f_n$: namely $\widehat{C}_n$ is a $(1 - \alpha)$-CI with respect to $f$ for all distributions $P$, and there exists a $P$ such that $P_{f_n(X)}$ is nonatomic. Thus Corollary 2 yields that we can construct a distribution $Q$ such that

$$\mathbb{E}_{Q^{n+1}}|C_n(f_n(X_{n+1}))| \geqslant 0.5 - \alpha,$$

which is a contradiction to (30). Hence our hypothesis that $\limsup_{n\to\infty}|\mathcal{X}^{(f_n)}| > \aleph_0$ must be false, concluding the proof. $\qquad\square$

# C  Proofs of results in Section 4 (other than Section 4.4)

## C.1  Proof of Theorem 4

Let $E_{\mathcal{B}(x)}$ be the event that $(\mathcal{B}(X_1), \ldots, \mathcal{B}(X_n)) = (\mathcal{B}(x_1), \ldots, \mathcal{B}(x_n))$. On the event $E_{\mathcal{B}(x)}$, within each region $\mathcal{X}_b$, the number of point from the calibration set is known and the $Y_i$'s in each bin represent independent Bernoulli random variables that share the same mean $\pi_b = \mathbb{E}\left[Y \mid X \in \mathcal{X}_b\right]$. Consider any fixed region $\mathcal{X}_b$, $b \in [B]$. Using Theorem 10, we obtain that:

$$\mathbb{P}\left(|\pi_b - \widehat{\pi}_b| > \sqrt{\frac{2\widehat{V}_b \ln(3B/\alpha)}{N_b}} + \frac{3\ln(3B/\alpha)}{N_b} \;\middle|\; E_{\mathcal{B}(x)}\right) \leqslant \alpha/B.$$

Applying union bound across all regions of the sample-space partition, we get that:

$$\mathbb{P}\left(\forall b \in [B]: \; |\pi_b - \widehat{\pi}_b| \leqslant \sqrt{\frac{2\widehat{V}_b \ln(3B/\alpha)}{N_b}} + \frac{3\ln(3B/\alpha)}{N_b} \;\middle|\; E_{\mathcal{B}(x)}\right) \geqslant 1 - \alpha.$$

Because this is true for any $E_{\mathcal{B}(x)}$, we can marginalize to obtain the assertion of the theorem in unconditional form. $\qquad\square$

## C.2  Proof of Corollary 4

We show a calibration guarantee by using Theorem 1. Consider the scoring function as $\mathcal{B}$ with $\mathcal{Z} = [B]$. Then by Theorem 4, $C : [B] \to \mathcal{I}$ given by

$$C(b) = \left[\widehat{\pi}_b - \left(\sqrt{\frac{2\widehat{V}_b \ln(3B/\alpha)}{N_b}} + \frac{3\ln(3B/\alpha)}{N_b}\right), \widehat{\pi}_b + \sqrt{\frac{2\widehat{V}_b \ln(3B/\alpha)}{N_b}} + \frac{3\ln(3B/\alpha)}{N_b}\right], \; b \in [B],$$

provides a $(1-\alpha)$-CI with respect to $\mathcal{B}$. Let $b^\star = \min_{b \in [B]} N_b$. To apply Theorem 4, we define

$$\varepsilon = \sup_{b \in [B]} |C(b)| / 2 = \sqrt{\frac{2\widehat{V}_{b^\star} \ln(3B/\alpha)}{N_{b^\star}}} + \frac{3\ln(3B/\alpha)}{N_{b^\star}},$$

and the mid-point function $m_C$ for $C$ is given by $m_C(b) = \widehat{\pi}_b$. Applying Theorem 1 gives the first part of the result.

Next, suppose some bin $b$ has $\mathbb{P}(\mathcal{B}(X) = b) = 0$. Then, a test point $X_{n+1}$ almost surely does not belong to the bin, and the bin can be ignored for our calibration guarantee. Thus without loss of generality, suppose every $b \in [B]$ satisfies

$$\mathbb{P}(\mathcal{B}(X) = b) > 0.$$

Let $\min_{b \in [B]} \mathbb{P}(\mathcal{B}(X) = b) = \tau > 0$. Then for a fixed number of samples $n$, any particular bin $b$, and any constant $\alpha \in (0, 1)$ we have by Hoeffding's inequality with probability $1 - \alpha/B$

$$N_b \geqslant n\tau - \sqrt{\frac{n \ln(B/\alpha)}{2}}.$$

Taking a union bound, we have with probability $1 - \alpha$, simultaneously for every $b \in [B]$,

$$N_b \geqslant n\tau - \sqrt{\frac{n \ln(B/\alpha)}{2}} = \Omega(n),$$

and in particular $N_{b^\star} = \Omega(n)$ where $b^\star = \arg\min_{b \in [B]} N_b$. Thus by the first part of this corollary, $f_n$ is $\varepsilon_n$ calibrated where $\varepsilon_n = O(\sqrt{n^{-1}}) = o(1)$. This concludes the proof. $\qquad\square$

## C.3 Proof of Theorem 5

Denote $|\mathcal{D}_{cal}^2| = n$. Let $p_j = \mathbb{P}(g(X) \in I_j)$ be the true probability that a random point falls into partition $\mathcal{X}_j$. Assume $c$ is such that we can use Lemma 11 to guarantee that with probability at least $1 - \alpha/2$, uniform mass binning scheme is 2-well-balanced. Hence, with probability at least $1 - \alpha/2$:

$$\frac{1}{2B} \leqslant p_j \leqslant \frac{2}{B}, \ \forall j \in [B]. \tag{31}$$

Moreover, by Hoeffding's inequality we get that for any fixed region of sample-space partition, with probability at least $1 - \alpha/2B$, for a fixed $j \in [B]$,

$$N_j \geqslant np_j - \sqrt{\frac{n \ln(2B/\alpha)}{2}}. \tag{32}$$

Hence, by union bound across applied accross all regions and using (31), we get that with probability at least $1 - \alpha/2$:

$$N_{b^\star} \geqslant \frac{n}{2B} - \sqrt{\frac{n \ln(2B/\alpha)}{2}},$$

where the first term dominates asymptotically (for fixed $B$). Hence, we get that with probability at least $1 - \alpha$, $N_{b^\star} = \Omega(n/B)$. By invoking the result of Corollary 4 and observing that $\widehat{V}_b \leqslant 1$, we conclude that uniform mass binning is $(\varepsilon, \alpha)$ approximately calibrated with $\varepsilon = O(\sqrt{B \ln(B/\alpha)/n})$ as desired. This also leads to asymptotic calibration by Corollary 4. $\qquad\square$

## C.4 Proof of Theorem 6

The proof is based on the result for an empirical-Bernstein confidence sequences for bounded observations [15]. We condition on the event $E_{\mathcal{B}(x)}^\infty$ defined as $(\mathcal{B}(X_1), \mathcal{B}(X_1), \dots) = (\mathcal{B}(x_1), \mathcal{B}(x_2), \dots)$, that is the random variables denoting which partition the infinite stream of samples fall in (thus allowing our bound to hold for every possible value of $n$). On $E_{\mathcal{B}(x)}^\infty$, the label values within each partition of the sample-space partition represent independent Bernoulli random variable that share the same mean $\pi_b = \mathbb{E}[Y \mid X \in \mathcal{X}_b], b \in [B]$. Consequently, the bound obtained can be marginalized over $E_{\mathcal{B}(x)}^\infty$ to obtain the assertion of the theorem in unconditional form. Now we show the bound that applies conditionally on $E_{\mathcal{B}(x)}^\infty$.

Consider any fixed region of the sample-space partition $\mathcal{X}_b$ and corresponding points $\{(X_i^b, Y_i^b)\}_{i=1}^{N_b}$. Then $S_t = \left(\sum_{i=1}^t Y_i^b\right) - t\pi_b$ is a sub-exponential process with variance process:

$$\widehat{V}_t^+ = \sum_{i=1}^t \left(Y_i^b - \overline{Y}_{i-1}^b\right)^2.$$

Howard et al. [14, Proposition 2] implies that $S_t$ is also a sub-gamma process with variance process $\widehat{V}_t$ and the same scale $c = 1$. Since the theorem holds for any sub-exponential uniform boundary, we choose one based on analytical convenience. Recall definition of the polynomial stitching function

$$\mathcal{S}_\alpha(v) := \sqrt{k_1^2 v l(v) + k_2^2 c^2 l^2(v)} + k_2 c l(v), \quad \text{where} \quad \begin{cases} l(v) := \ln h(\ln_\eta(v/m)) + \ln(l_0/\alpha), \\ k_1 := (\eta^{1/4} + \eta^{-1/4})/\sqrt{2}, \\ k_2 := (\sqrt{\eta} + 1)/\sqrt{2}. \end{cases}$$

where $l_0 = 1$ for the scalar case. Note that for $c > 0$ it holds that $\mathcal{S}_\alpha(v) \leqslant k_1\sqrt{v l(v)} + 2ck_2 l(v)$.

From Howard et al. [15, Theorem 1], it follows that $u(v) = \mathcal{S}_\alpha(v \vee m)$ is a sub-gamma uniform boundary with scale $c$ and crossing probability $\alpha$. Applying Theorem 9 with $h(k) \leftarrow (k+1)^s \zeta(s)$ where $\zeta(\cdot)$ is Riemann zeta function and parameters $\eta \leftarrow e$, $s \leftarrow 1.4$, $c \leftarrow 1$, $m \leftarrow 1$ and $\alpha \leftarrow \alpha/(2B)$, yields that $k_2 \leqslant 1.88, k_1 \leqslant 1.46$ and $l(v) = 1.4 \cdot \ln\ln(ev) + \ln(2\zeta(1.4)B/\alpha)$. Since Theorem 9 provides a bound that holds uniformly across time $t$, then it provides a guarantee for $t = N_b$, in particular. Hence, with probability at least $1 - \alpha/B$,

$$|\pi_b - \widehat{\pi}_b| \leqslant \frac{1.46\sqrt{\widehat{V}_b^+ \cdot 1.4 \cdot \ln\ln\left(e\left(\widehat{V}_b^+ \vee 1\right)\right) + \ln(6.3B/\alpha)}}{N_b} + \frac{5.27 \cdot \ln\ln\left(e\left(\widehat{V}_b^+ \vee 1\right)\right) + 3.76\ln(6.3B/\alpha)}{N_b}$$

$$\leqslant \frac{7\sqrt{\widehat{V}_b^+ \cdot \ln\ln\left(e\left(\widehat{V}_b^+ \vee 1\right)\right)} + 5.3\ln(6.3B/\alpha)}{N_b}.$$

using that $\sqrt{x+y} \leqslant \sqrt{x} + \sqrt{y}$ and $\ln\ln(ex) \leqslant \sqrt{x \ln\ln ex}$ for $x \geqslant 1$. Finally, we apply a union bound to get a guarantee that holds simultaneously for all regions of the sample-space partition. $\qquad\square$

# D   Calibration under covariate shift (including proofs of results in Section 4.4)

The results from Section 4.4 are proved in Appendix D.1 (Theorem 7) and D.3 (Proposition 2). To show Theorem 7, we first propose and analyze a slightly different estimator than (39) that is unbiased for $\pi_b^{(w)}$, but needs additional oracle access to the parameters $\{m_b\}_{b\in[B]}$ defined as

$$m_b = \mathbb{P}_{P_X}(X \in \mathcal{X}_b) \,/\, \mathbb{P}_{\widetilde{P}_X}(X \in \mathcal{X}_b).$$

$m_b$ denotes the 'relative mass' of region $\mathcal{X}_b$. (For simplicity, we assume that $\mathbb{P}_{\widetilde{P}}(X \in \mathcal{X}_b) > 0$ for every $b$ since otherwise the test-point almost surely does not belong to $\mathcal{X}_b$ and estimation in that bin is not relevant for a calibration guarantee.) We then show that $m_b$ can be estimated using $w$, which would lead to the proposed estimator $\breve{\pi}_b^{(w)}$. First, we establish the following relationship between $\mathbb{E}_{\widetilde{P}}\left[Y \mid X \in \mathcal{X}_b\right]$ and $\mathbb{E}_P\left[Y \mid X \in \mathcal{X}_b\right]$.

**Proposition 3.** *Under the covariate shift assumption, for any $b \in [B]$*

$$\mathbb{E}_{\widetilde{P}}\left[Y \mid X \in \mathcal{X}_b\right] = m_b \cdot \mathbb{E}_P\left[w(X)Y \mid X \in \mathcal{X}_b\right].$$

*Proof.* Observe that

$$\frac{d\widetilde{P}(X \mid X \in \mathcal{X}_b)}{dP(X \mid X \in \mathcal{X}_b)} = \frac{d\widetilde{P}(X)}{dP(X)} \cdot \frac{\mathbb{P}_P\left(X \in \mathcal{X}_b\right)}{\mathbb{P}_{\widetilde{P}}(X \in \mathcal{X}_b)} = w(X) \cdot m_b.$$

Thus we have,

$$
\begin{aligned}
\mathbb{E}_{\widetilde{P}}\left[Y \mid X \in \mathcal{X}_b\right] &\overset{(1)}{=} \mathbb{E}_{\widetilde{P}}\left[\mathbb{E}_{\widetilde{P}}\left[Y \mid X\right] \mid X \in \mathcal{X}_b\right] \\
&\overset{(2)}{=} \mathbb{E}_{\widetilde{P}}\left[\mathbb{E}_P\left[Y \mid X\right] \mid X \in \mathcal{X}_b\right] \\
&\overset{(3)}{=} \mathbb{E}_P\left[\frac{d\widetilde{P}(X \mid X \in \mathcal{X}_b)}{dP(X \mid X \in \mathcal{X}_b)} \cdot \mathbb{E}_P\left[Y \mid X\right] \mid X \in \mathcal{X}_b\right] \\
&\overset{(4)}{=} m_b \cdot \mathbb{E}_P\left[w(X)\mathbb{E}_P\left[Y \mid X\right] \mid X \in \mathcal{X}_b\right] \\
&\overset{(5)}{=} m_b \cdot \mathbb{E}_P\left[\mathbb{E}_P\left[w(X)Y \mid X\right] \mid X \in \mathcal{X}_b\right] \\
&\overset{(6)}{=} m_b \cdot \mathbb{E}_P\left[w(X)Y \mid X \in \mathcal{X}_b\right],
\end{aligned}
$$

where in (1) we use the tower rule, in (2) we use the covariate shift assumption, (3) can be seen by using the integral form of the expectation, (4) uses the observation at the beginning of the proof, (5) uses that $w(X)$ is a function of $X$ and finally, (6) uses the tower rule. $\qquad\square$

Let $N_b$ denote the number of calibration points from the source domain that belong to bin $b$. Given Proposition 3, a natural estimator for $\mathbb{E}_{\widetilde{P}}\left[Y \mid X \in \mathcal{X}_b\right]$ is given by:

$$\widehat{\pi}_b^{(w)} := \frac{1}{N_b}\sum_{i:\mathcal{B}(X_i)=b} m_b w(X_i)Y_i. \tag{33}$$

Estimation properties of $\widehat{\pi}_b^{(w)}$ are given by the following theorem.

**Theorem 8.** *Assume that $\sup_x w(x) = U < \infty$. For any $\alpha \in (0,1)$, with probability at least $1-\alpha$,*

$$\left| \widehat{\pi}_b^{(w)} - \mathbb{E}_{\widetilde{P}}\left[ Y \mid X \in \mathcal{X}_b \right] \right| \leqslant \sqrt{\frac{2\widehat{V}_b^{(w)} \ln(3B/\alpha)}{N_b}} + \frac{3 m_b U \ln(3B/\alpha)}{N_b}, \quad \textit{simultaneously for all } b \in [B],$$

*where $\widehat{V}_b^{(w)} = \frac{1}{N_b} \sum_{i:\mathcal{B}(X_i)=b}(m_b w(X_i) Y_i - \widehat{\pi}_b^{(w)})^2$.*

The proof is given in Appendix D.2. Next, we discuss a way of estimating $m_b$ using likelihood ratio $w$ instead of relying on oracle access. Observe that

$$\frac{d\widetilde{P}(X \mid X \in \mathcal{X}_b)}{dP(X \mid X \in \mathcal{X}_b)} = \frac{d\widetilde{P}(X)}{dP(X)} \cdot \frac{\mathbb{P}_P\left(X \in \mathcal{X}_b\right)}{\mathbb{P}_{\widetilde{P}}(X \in \mathcal{X}_b)} = w(X) \cdot m_b.$$

Thus we have,

$$\mathbb{E}_P\left[ w(X) \mid X \in \mathcal{X}_b \right] = m_b^{-1}\mathbb{E}_P\left[ \frac{d\widetilde{P}(X \mid X \in \mathcal{X}_b)}{dP(X \mid X \in \mathcal{X}_b)} \mid X \in \mathcal{X}_b \right] = m_b^{-1}, \qquad (34)$$

which suggests a possible estimator for $m_b$ given by

$$\widehat{m}_b = \left( \frac{\sum_{i:\mathcal{B}(X_i)=b} w(X_i)}{N_b} \right)^{-1}, \quad b \in [B]. \qquad (35)$$

On substituting this estimate for $m_b$ in (33), we get a new estimator

$$\frac{\sum_{i:\mathcal{B}(X_i)=b} w(X_i) Y_i}{\sum_{i:\mathcal{B}(X_i)=b} w(X_i)},$$

which is exactly $\breve{\pi}_b^{(w)}$. With this observation, we now prove Theorem 7.

### D.1 Proof of Theorem 7

Let us define $r_b := 1/m_b$ and

$$\widehat{r}_b = \frac{\sum_{i:\mathcal{B}(X_i)=b} w(X_i)}{N_b}. \qquad (36)$$

**Step 1 (Uniform lower bound for $N_b$).** Since the regions of the sample-space partition were constructed using uniform-mass binning, the guarantee of Theorem 5 holds. Precisely, we have that with probability at least $1 - \alpha/3$, simultaneously for every $b \in [B]$,

$$N_b \geqslant \frac{n}{2B} - \sqrt{\frac{n \ln(6B/\alpha)}{2}}.$$

**Step 2 (Approximating $r_b$).** Observe that the estimator (36) is an average of $N_b$ random variables bounded by the interval $[0, U]$. Let $E_{\mathcal{B}(x)}$ be the event that $(\mathcal{B}(X_1), \ldots, \mathcal{B}(X_n)) = (\mathcal{B}(x_1), \ldots, \mathcal{B}(x_n))$. On the event $E_{\mathcal{B}(x)}$, within each region $\mathcal{X}_b$, the number of point from the calibration set is known and the $Y_i$'s in each bin represent independent Bernoulli random variables that share the same mean $\mathbb{E}\left[ w(X) \mid X \in \mathcal{X}_b \right]$. Consider any fixed region $\mathcal{X}_b$, $b \in [B]$. By Hoeffding's inequality, it holds that

$$\mathbb{P}\left( |r_b - \widehat{r}_b| > \sqrt{\frac{U^2 \ln(6B/\alpha)}{2N_b}} \mid E_{\mathcal{B}(x)} \right) \leqslant \alpha/(3B).$$

Applying union bound across all regions of the sample-space partition, we get that:

$$\mathbb{P}\left( \exists b \in [B]: \ |r_b - \widehat{r}_b| > \sqrt{\frac{U^2 \ln(6B/\alpha)}{2N_b}} \mid E_{\mathcal{B}(x)} \right) \leqslant \alpha/3.$$

Because this is true for any $E_{\mathcal{B}(x)}$, we can marginalize to obtain that with probability at least $1 - \alpha/3$,

$$\forall b \in [B], \ |r_b - \widehat{r}_b| \leqslant \sqrt{\frac{U^2 \ln(6B/\alpha)}{2N_b}}. \qquad (37)$$

**Step 3 (Going from $r_b$ to $m_b$).** Define $r^\star = \min_{b \in [B]} \mathbb{E}\left[w(X) \mid X \in \mathcal{X}_b\right]$. Suppose $\forall b \in [B]$, $|r_b - \widehat{r}_b| \leqslant \varepsilon$ and $\varepsilon < r^\star/2$. Then, we have with probability at least $1 - \alpha/3$:

$$|m_b - \widehat{m}_b| = \left| \frac{1}{r_b} - \frac{1}{\widehat{r}_b} \right| = \left| \frac{r_b - \widehat{r}_b}{r_b \cdot \widehat{r}_b} \right| \leqslant \frac{\varepsilon}{r_b^2 |1 - \varepsilon/r_b|} \leqslant \frac{2\varepsilon}{r_b^2} = 2m_b^2 \varepsilon, \quad \forall b \in [B]. \qquad (38)$$

We now set $\varepsilon = \sqrt{\frac{U^2 \ln(6B/\alpha)}{2N_b}}$ as specified in equation (37) and verify that $\varepsilon < r^\star/2$.

- First, from step 1, with probability at least $1 - \alpha/3$, $N_{b^\star} = \Omega(n/B)$ and thus $N_b = \Omega(n/B)$ for every $b \in [B]$.

- By the condition in the theorem statement, for every $b \in [B]$,

$$\varepsilon = \sqrt{\frac{U^2 \ln(6B/\alpha)}{2N_b}} = O\left( \sqrt{\frac{U^2 B \ln(6B/\alpha)}{n}} \right) = O\left( \sqrt{\frac{U^2 B \ln(6B/\alpha)}{\left( \frac{U^2 B \ln(6B/\alpha)}{L^2} \right)}} \right) = O\left( L \right).$$

  Finally recall that $L \leqslant r^\star$. Thus we can pick $c$ in the theorem statement to be large enough such that $\varepsilon < L/2 \leqslant r^\star/2$.

Thus for $\varepsilon = \sqrt{\frac{U^2 \ln(6B/\alpha)}{2N_b}}$, by a union bound over the event in (37) and step 1, the conditions for (38) are satisfied with probability at least $1 - 2\alpha/3$. Hence we have for some large enough constant $c > 0$,

$$|m_b - \widehat{m}_b| \leqslant cm_b^2 \cdot \sqrt{\frac{U^2 B \ln(6B/\alpha)}{2n}} \leqslant c \cdot \frac{U}{L^2} \sqrt{\frac{B \ln(6B/\alpha)}{2n}}.$$

The final inequality holds by observing that $m_b \leqslant 1/L$ which follows from relationship (34) and the assumption that $\inf_x w(x) \geqslant L$.

**Step 4 (Computing the final deviation inequality for $\breve{\pi}_b^{(w)}$).** Recall the definitions of the two estimators:

$$\widehat{\pi}_b^{(w)} := \frac{1}{N_b} \sum_{i:\mathcal{B}(X_i)=b} m_b w(X_i) Y_i,$$

and

$$\breve{\pi}_b^{(w)} := \frac{1}{N_b} \sum_{i:\mathcal{B}(X_i)=b} \widehat{m}_b w(X_i) Y_i,$$

which differ by replacing $m_b$ by its estimator $\widehat{m}_b$ defined in (35). By triangle inequality,

$$\left| \breve{\pi}_b - \mathbb{E}\left[Y \mid X \in \mathcal{X}_b\right] \right| \leqslant \left| \breve{\pi}_b^{(w)} - \widehat{\pi}_b^{(w)} \right| + \left| \widehat{\pi}_b^{(w)} - \mathbb{E}\left[Y \mid X \in \mathcal{X}_b\right] \right|.$$

Theorem 8 bounds the term $\left| \widehat{\pi}_b^{(w)} - \mathbb{E}\left[Y \mid X \in \mathcal{X}_b\right] \right|$ with high probability. In the proof of Theorem 8, we can replace the empirical Bernstein's inequality by Hoeffding's inequality to obtain with probability at least $1 - \alpha/3$,

$$\left| \widehat{\pi}_b^{(w)} - \mathbb{E}\left[Y \mid X \in \mathcal{X}_b\right] \right| \leqslant \sqrt{\frac{U^2 \ln(6B/\alpha)}{2N_b}} \leqslant \left( \frac{U}{L} \right)^2 \sqrt{\frac{\ln(6B/\alpha)}{2N_b}},$$

simultaneously for all $b \in [B]$ (the last inequality follows since $L \leqslant 1 \leqslant U$). To bound $\left| \widehat{\pi}_b^{(w)} - \breve{\pi}_b^{(w)} \right|$, first note that:

$$\left| \widehat{\pi}_b^{(w)} - \breve{\pi}_b^{(w)} \right| = \left| \frac{1}{N_b} \sum_{i:\mathcal{B}(X_i)=b} (\widehat{m}_b - m_b) w(X_i) Y_i \right|$$

$$\leqslant U \cdot \left| \frac{1}{N_b} \sum_{i:\mathcal{B}(X_i)=b} (\widehat{m}_b - m_b) \right|$$

$$= U \cdot |\widehat{m}_b - m_b|.$$

Then we use the results from steps 1 and 3 to conclude that with probability at least $1 - 2\alpha/3$,

$$\left|\breve{\pi}_b^{(w)} - \widehat{\pi}_b^{(w)}\right| \leqslant c \cdot \left(\frac{U}{L}\right)^2 \sqrt{\frac{B \ln(6B/\alpha)}{2n}}, \quad \text{and} \quad N_b \geqslant n/B - \sqrt{\frac{n \ln(6B/\alpha)}{2}}.$$

simultaneously for all $b \in [B]$. Thus by union bound, we get that it holds with probability at least $1 - \alpha$,

$$|\breve{\pi}_b - \mathbb{E}\left[Y \mid X \in \mathcal{X}_b\right]| \leqslant c \cdot \left(\frac{U}{L}\right)^2 \sqrt{\frac{B \ln(6B/\alpha)}{2n}},$$

simultaneously for all $b \in [B]$ and large enough absolute constant $c > 0$. This concludes the proof. $\square$

## D.2 Proof of Theorem 8

Consider the event $E_{\mathcal{B}(x)}$ defined as $(\mathcal{B}(X_1), \ldots, \mathcal{B}(X_n)) = (\mathcal{B}(x_1), \ldots, \mathcal{B}(x_n))$. Conditioned on $E_{\mathcal{B}(x)}$, since $\sup_x w(x) \leqslant U$, we get that $\widehat{\pi}_b^{(w)}$ is an average of independent non-negative random variables $m_b w(X_i) Y_i$ that are bounded by $m_b U$ and share the same mean $m_b \mathbb{E}_P\left[w(X)Y \mid X \in \mathcal{X}_b\right] = \mathbb{E}_{\widetilde{P}}\left[Y \mid X \in \mathcal{X}_b\right]$ (by Proposition 3).Using Theorem 10 for a fixed $b \in [B]$, we obtain:

$$\mathbb{P}\left(\left|\widehat{\pi}_b^{(w)} - \mathbb{E}_{\widetilde{P}}\left[Y \mid X \in \mathcal{X}_b\right]\right| > \sqrt{\frac{2\widehat{V}_b \ln(3B/\alpha)}{N_b}} + \frac{3m_b U \ln(3B/\alpha)}{N_b} \,\Big|\, E_{\mathcal{B}(x)}\right) \leqslant \alpha/B.$$

Applying a union bound over all $b \in [B]$, we get:

$$\mathbb{P}\left(\forall b \in [B]: \left|\widehat{\pi}_b^{(w)} - \mathbb{E}_{\widetilde{P}}\left[Y \mid X \in \mathcal{X}_b\right]\right| \leqslant \sqrt{\frac{2\widehat{V}_b \ln(3B/\alpha)}{N_b}} + \frac{3m_b U \ln(3B/\alpha)}{N_b} \,\Big|\, E_{\mathcal{B}(x)}\right) \geqslant 1 - \alpha.$$

Because this is true for any $E_{\mathcal{B}(x)}$, we can marginalize to obtain the assertion of the theorem in unconditional form. $\square$

## D.3 Proof of Proposition 2

Fix any $\alpha \in (0, 1)$. For any $k \in \mathbb{N}$ observe that by triangle inequality,

$$\left|\breve{\pi}_b^{(\widehat{w}_k)} - \mathbb{E}_{\widetilde{P}}\left[Y \mid X \in \mathcal{X}_b\right]\right| \leqslant \left|\breve{\pi}_b^{(w)} - \mathbb{E}_{\widetilde{P}}\left[Y \mid X \in \mathcal{X}_b\right]\right| + \left|\breve{\pi}_b^{(w)} - \breve{\pi}_b^{(\widehat{w}_k)}\right|.$$

Consider any $\varepsilon > 0$. Note that by Theorem 7, there exists sufficiently large $n$ such that the first term is larger than $\varepsilon/2$ with probability at most $\alpha/2$ simultaneously for all $b \in [B]$. Hence, it suffices to show that there exists a large enough $k$ such that the probability of the second term exceeding $\varepsilon/2$ is at most $\alpha/2$ simultaneously for all $b \in [B]$. While analyzing the second term, we treat $n$ as a constant while leveraging the consistency of $\widehat{w}_k$ as $k \to \infty$. For simplicity, denote $\Delta_k = \sup_x |w(x) - \widehat{w}_k(x)|$. Then for any $b \in [B]$:

$$
\begin{aligned}
\left|\breve{\pi}_b^{(w)} - \breve{\pi}_b^{(\widehat{w}_k)}\right| &= \left|\frac{\sum_{i:\mathcal{B}(X_i)=b} w(X_i) Y_i}{\sum_{i:\mathcal{B}(X_i)=b} w(X_i)} - \frac{\sum_{i:\mathcal{B}(X_i)=b} \widehat{w}_k(X_i) Y_i}{\sum_{i:\mathcal{B}(X_i)=b} \widehat{w}_k(X_i)}\right| \\
&\overset{(1)}{\leqslant} \left|\frac{\sum_{i:\mathcal{B}(X_i)=b} w(X_i) Y_i}{\sum_{i:\mathcal{B}(X_i)=b} w(X_i)} - \frac{\sum_{i:\mathcal{B}(X_i)=b} \widehat{w}_k(X_i) Y_i}{\sum_{i:\mathcal{B}(X_i)=b} w(X_i)}\right| \\
&\quad + \left|\frac{\sum_{i:\mathcal{B}(X_i)=b} \widehat{w}_k(X_i) Y_i}{\sum_{i:\mathcal{B}(X_i)=b} w(X_i)} - \frac{\sum_{i:\mathcal{B}(X_i)=b} \widehat{w}_k(X_i) Y_i}{\sum_{i:\mathcal{B}(X_i)=b} \widehat{w}_k(X_i)}\right| \\
&\overset{(2)}{\leqslant} n \cdot \Delta_k \cdot \left|\frac{1}{\sum_{i:\mathcal{B}(X_i)=b} w(X_i)}\right|
\end{aligned}
$$

$$+\left|\frac{1}{\sum_{i:\mathcal{B}(X_i)=b} w(X_i)} - \frac{1}{\sum_{i:\mathcal{B}(X_i)=b} \widehat{w}_k(X_i)}\right|\left|\sum_{i:\mathcal{B}(X_i)=b} \widehat{w}_k(X_i)Y_i\right|$$

$$\overset{(3)}{\leqslant} \frac{n}{L}\cdot\Delta_k + \left(\frac{n\cdot\Delta_k}{(L-\Delta_k)L}\right)\cdot\left((U+\Delta_k)\cdot n\right),$$

where (1) is due to the triangle inequality, (2) is due to the facts that the number of points in any bin is at most $n$ and that absolute difference between $\widehat{w}$ and $w$ is at most $\Delta_k$, (3) combines the aforementioned reasons in (2) and the assumptions: $L \leqslant \inf_x w(x) \leqslant \sup_x w(x) \leqslant U$. Since $\Delta_k \overset{P}{\to} 0$, clearly there exists a large enough $k$ such that:

$$\mathbb{P}\left(\left|\breve{\pi}_b^{(w)} - \breve{\pi}_b^{(\widehat{w}_k)}\right| \geqslant \varepsilon/2\right) \leqslant \alpha/2.$$

Thus we conclude that $\breve{\pi}_b^{(\widehat{w}_k)}$ is asymptotically calibrated at level $\alpha$. $\qquad\square$

### D.4 Preliminary simulations

This section is structured as follows. We first describe the overall procedure for calibration under covariate shift. The finite-sample calibration guarantee of Theorem 7 holds for oracle $w$ whereas in our experiments we will estimate $w$; to assess the loss in calibration due to this approximation, we introduce some standard techniques used in literature. The preliminary experiments are performed with simulated data which are described after this. Finally, we propose a modified estimator $\widetilde{\pi}_b^{(\widehat{w})}$ of $\mathbb{E}_{\widetilde{P}}\left[Y \mid X \in \mathcal{X}_b\right]$ which appears natural but has poor performance in practice.

**Procedure.** We describe how to construct approximately calibrated predictions practically. This involves approximating the importance weights $w$ and the relatives mass terms $\{m_b\}_{b\in[B]}$. The summarized calibration procedure consists of the following steps:

1. Split the calibration set into two parts and use the first to perform *uniform mass* binning
2. Given unlabeled examples from both source and target domain, estimate $\widehat{w}$. The unconstrained Least-Squares Importance Fitting (uLSIF) procedure [17] is used for this.
3. Compute for every $b \in [B]$, the estimator as per (17), replacing $w$ with $\widehat{w}$:

$$\breve{\pi}_b^{(\widehat{w})} := \frac{\sum_{i:\mathcal{B}(X_i)=b} \widehat{w}(X_i)Y_i}{\sum_{i:\mathcal{B}(X_i)=b} \widehat{w}(X_i)}. \tag{39}$$

4. On a new test point from the target distribution, output the calibrated estimate $\breve{\pi}_{\mathcal{B}(X_{n+1})}^{(\widehat{w})}$.

**Assessment through reliability diagrams and ECE.** Given a test set (from the target distribution) of size $m$: $\{(X_i', Y_i')\}_{i\in[m]}$ and a function $g : \mathcal{X} \to [0,1]$ that outputs approximately calibrated probabilities, we consider the reliability diagram to estimate its calibration properties. A reliability diagram is constructed using splitting the unit interval $[0,1]$ into non-overlapping intervals $\{I_b\}_{b\in[B']}$ for some $B'$ as

$$I_i = \left[\frac{i-1}{B'}, \frac{i}{B'}\right), \ i = 1,\dots,B'-1 \ \text{and} \ I_{B'} = \left[\frac{B'-1}{B'}, 1\right].$$

Let $\mathcal{B}' : [0,1] \to [B']$ denote the binning function that corresponds to this binning. We then compute the following quantities for each bin $b \in [B']$:

$$\text{FP}(I_b) = \frac{\sum_{i:\mathcal{B}'(X_i')=b} Y_i'}{|\{i : \mathcal{B}'(X_i') = b\}|} \qquad \text{(fraction of positives in a bin),}$$

$$\text{MP}(I_b) = \frac{\sum_{i:\mathcal{B}'(X_i')=b} g(X_i')}{|\{i : \mathcal{B}'(X_i') = b\}|} \qquad \text{(mean predicted probability in a bin).}$$

If $g$ is perfectly calibrated, the reliability diagram is diagonal. Define the proportion of points that fall into various bins as:

$$\widehat{p}_b = \frac{|\{i : \mathcal{B}'(X_i') = b\}|}{m}, \quad b \in [B'].$$

Figure 2: In Figure 2a uncalibrated Random Forest (ECE $\approx 0.023$) is compared with calibration that does not take the covariate shift into account (ECE $\approx 0.047$). In Figure 2b uncalibrated Random Forest is compared with calibration that takes the covariate shift into account (ECE $\approx 0.017$).

Then ECE (or $\ell_1$-ECE) is defined as:

$$\text{ECE}(g) = \sum_{b \in [B']} \widehat{p}_b \cdot |\text{MP}(I_b) - \text{FP}(I_b)| .$$

ECE can also be defined in the $\ell_p$ sense and for multiclass problems but we limit our attention to the $\ell_1$-ECE for binary problems.

**Simulations with synthetic data.** We illustrate the performance of our proposed estimator (17) using the following simulated example, for which we can explicitly control the covariate shift. Consider the following data generation pipeline: for the source domain each component of the feature vector is drawn from Beta$(\alpha, \beta)$ where $\alpha = \beta = 1$, which corresponds to uniform draws from the unit cube. For the target distribution each component can be drawn independently from Beta$(\alpha', \beta')$. If the dimension is $d$, the true likelihood ratio is given as

$$w(x) = \frac{d\widetilde{P}_X(x)}{dP_X(x)} = \frac{B^d(\alpha; \beta)}{B^d(\alpha'; \beta')} \prod_{i=1}^{d} \frac{(x_{(i)})^{\alpha'-1}(1 - x_{(i)})^{\beta'-1}}{(x_{(i)})^{\alpha-1}(1 - x_{(i)})^{\beta-1}},$$

where $x_{(i)}$ are the coordinates of feature vector $x$. We set $d = 3$ and $\alpha' = 2, \beta' = 1$ so that $w(x) = 8 \cdot x_{(1)} x_{(2)} x_{(3)}$. The labels for both source and target distributions are assigned according to:

$$\mathbb{P}(Y = 1 \mid X = x) = \frac{1}{2} \left(1 + \sin\left(\omega \left(x_{(1)}^2 + x_{(2)}^2 + x_{(3)}^2\right)\right)\right),$$

for $\omega = 20$. As the underlying classifier we use a Random Forest with 100 trees (from `sklearn`). 14700 data points were used to train the underlying Random Forest classifier, 2000 data points from both source and target were used for the estimation of importance weights. The parameters $\sigma$ and $\lambda$ for uLSIF were tuned by leave-one-out cross-validation: we considered 25 equally spaced values on a log-scale in range $(10^{-2}, 10^2)$ for $\sigma$ and 100 equally spaced values on a log-scale in range $(10^{-3}, 10^3)$ for $\lambda$. Uniform mass binning was performed with 10 bins and 1940 data points from the source domain were used to estimate the quantiles. 7840 source data points were used for the calibration and finally, 28000 data points from the target domain were used for evaluation purposes. We note that this simulation is a 'proof-of-concept'; the sample sizes we used are not necessarily optimal can presumably be improved.

We compare the unweighted estimator (12) which corresponds to weighing points in each bin equally as we would do if there was no covariate shift, and the estimator (17) that uses an estimate of $w$ to account for covariate shift. The reliability diagrams are presented in Figure 2, with the ECE reported in the caption. For the ECE estimation and reliability diagrams, we used $B' = 10$.

Figure 3: Calibration of Random Forest with $m_b$ estimated as per equation (35) (ECE $\approx 0.05$).

**Alternative estimator for** $m_b$. Estimator (35) is one way of estimating $m_b$ using the $w$ values, that leads to (17). However, there exists another natural estimator which we propose and show some preliminary empirical results for. Suppose we have access to additional unlabeled data from the source and target domains ($\{X_i^s\}_{i \in [n_s]}$, and $\{X_i^t\}_{i \in [n_t]}$ respectively). From the definition of $m_b = \mathbb{P}_{P_X}(X \in \mathcal{X}_b)/\mathbb{P}_{\tilde{P}_X}(X \in \mathcal{X}_b)$, a natural estimator is,

$$\widehat{m}_b = \frac{\frac{1}{n_s}|\{i \in [n_s] : \mathcal{B}(X_i^s) = b\}|}{\frac{1}{n_t}|\{i \in [n_t] : \mathcal{B}(X_i^t) = b\}|}, \quad b \in [B]. \tag{40}$$

In this case, the estimator (33) reduces to:

$$\widetilde{\pi}_b^{(\widehat{w})} = \frac{\widehat{m}_b}{N_b} \sum_{i:\mathcal{B}(X_i)=b} \widehat{w}(X_i) Y_i.$$

We show experimental results with this estimation procedure. We used 8500 data points from the source domain and 8000 points from the target domain to compute (40). The reliability diagram and ECE with this estimator is reported in Figure 3. On our simulated dataset, we observe that the estimators $\widetilde{\pi}_b^{(\widehat{w})}$ perform significantly worse than the estimators $\widecheck{\pi}_b^{(\widehat{w})}$. While this is only a single experimental setup, we outline some drawbacks of this estimation method that may lead to poor performance in general.

1. $\widetilde{\pi}_b^{(\widehat{w})}$ requires access to additional unlabeled data from the source and target domains without leading to increase in performance.

2. The denominator of $\widehat{m}_b$ could be badly behaved if the number of points from the target domain in bin $b$ are small. We could perform uniform-mass binning on the target domain to avoid this, but in this case $N_b$ may be small which would lead to the estimator $\widetilde{\pi}_b^{(\widehat{w})}$ performing poorly.

Our overall recommendation through these preliminary experiments is to use the estimator $\widehat{\pi}_b^{(\widehat{w})}$ as proposed in Section 4.4 instead of $\widetilde{\pi}_b^{(\widehat{w})}$.

## E   Venn prediction

Venn prediction [24, 45–47] is a calibration framework that provides distribution-free guarantees, which are different from the ones in Definitions 1 and 2. For a multiclass problem with $L$ labels, Venn prediction produces $L$ predictions, one of which is guaranteed to be perfectly calibrated

(although it is impossible to know which one). These are called multiprobabilistic predictors, formally defined as a collection of predictions $(f_1, f_2, \ldots f_L)$ where each $f_i \in \{\mathcal{X} \to \Delta_{L-1}\}$ (here $\Delta_{L-1}$ is the boundary of the $\ell_1$ ball in the non-negative orthant of $\mathbb{R}^L$, corresponding to all possible distributions over $\{1, 2, \ldots, L\}$). Vovk and Petej [45] defined two calibration guarantees for multiprobabilistic predictors, the first being oracle calibration.

**Definition 4** (Oracle calibration). $(f_1, f_2, \ldots f_L)$ is oracle calibrated if there exists an oracle selector $S$ such that $f_S$ is perfectly calibrated.

Venn predictors satisfy oracle calibration [45, Theorem 1] with $S = Y$. In the binary case, this means that when $Y = 1$, $f_1(X)$ is perfectly calibrated but we do not have any guarantee on $f_0(X)$; on the other hand if $Y = 0$, $f_0(X)$ is perfectly calibrated but we know nothing about $f_1(X)$. Since $Y$ is unknown, oracle calibration seems to us to primarily serve as theoretical guidance, but does not give a clear prescription on what to output and what theoretical guarantee that output satisfies. In practice, it seems reasonable to suspect that if $f_0(X)$ and $f_1(X)$ are close, then their average should be approximately calibrated in the sense of Definition 1, but to the best of our knowledge, such results have not been shown formally (other aggregate functions apart from average are also suggested (without formal guarantees) by Vovk and Petej [45, Section 4]). For instance, it may be tempting to think that oracle calibration of a multiprobabilistic predictor leads to approximate calibration in the following way. Consider the prediction function

$$f(X) = \frac{\min f_i(X) + \max f_i(X)}{2},$$

and the radius of the interval $[\min f_i(X), \max f_i(X)]$:

$$\varepsilon(X) = \frac{\max f_i(X) - \min f_i(X)}{2}.$$

Since Venn predictors satisfy oracle calibration, one might conjecture that $f$ is $(\varepsilon, \alpha)$ approximately calibration (per Definition 1) for the given function $\varepsilon$ and for any $\alpha \in (0, 1)$. We examined this claim but were unable to prove such a guarantee formally. In fact, it seems that no general calibration guarantee should be possible with the size of the calibration interval being $O(\varepsilon(X))$; we evidence this through the following construction.

Consider a setup, with no covariates and only label values $Y$, and a single bin that contains all points (in the Venn prediction language: a taxonomy under which all points are equivalent). For a test-point $Y_{n+1}$ and any predictor $f$, note that $\mathbb{E}[Y_{n+1} \mid f]$ is simply equal to $\mathbb{E}[Y_{n+1}]$ since any information used to construct $f$ is independent of $Y_{n+1}$. To ensure calibration, we may look for a guarantee of the following form for some $\delta$:

$$|\mathbb{E}[Y_{n+1} \mid f] - f| = |\mathbb{E}[Y_{n+1}] - f| \leqslant \delta.$$

In essence, $f$ is an estimator for the parameter $\mathbb{E}[Y]$ with a corresponding deviation bound of $\delta$. Without distributional assumptions, we only expect to estimate such a parameter with error at best $\delta = O(1/\sqrt{n})$ for a fixed constant probability of failure. On the other hand, the Venn prediction interval $[\min f_i, \max f_i]$ often has radius $O(1/n)$. Thus for valid approximate calibration, we would need to provide a larger interval than $[\min f_i, \max f_i]$, even though one of the $f_i$'s is perfectly calibrated. Given this example, our conjecture is that it might be possible to show that there always exists an $f_i(X)$ that is $(n^{-0.5}\text{polylog}(1/\alpha)), \alpha)$ calibrated. Without knowing which $f_i(X)$ to pick, perhaps one can show that an aggregate point in the interval $[\min f_i, \max f_i]$ is $((\max f_i - \min f_i) + n^{-0.5}\text{polylog}(1/\alpha), \alpha)$ approximately calibrated. In Section 4, we showed such a result for histogram binning (which can be interpreted as a Venn predictor). It would be interesting to study if such results can be shown for general Venn predictors.

Another guarantee for multiprobabilistic predictors is calibration in the large.

**Definition 5** (Calibration in the large). $(f_1, f_2, \ldots f_L)$ is calibrated in the large if the following is satisfied: $\mathbb{E}[Y] \in [\mathbb{E}\min f_i(X), \mathbb{E}\max f_i(X)]$.

Vovk and Petej [45, Theorem 2] show that Venn predictors satisfy calibration in the large. Due to the expectation signs and the coverage of the marginal probability $\mathbb{E}[Y]$, calibration in the large does not lead to a clear interpretable guarantee for uncertainty quantification, but rather a minimum requirement that serves as a guiding principle.

# F  Auxiliary results

## F.1  Concentration inequalities

**Theorem 9** (Howard et al. [15], Theorem 4). *Suppose $Z_t \in [a, b]$ a.s. for all $t$. Let $(\widehat{Z}_t)$ be any $[a, b]$-valued predictable sequence, and let $u$ be any sub-exponential uniform boundary with crossing probability $\alpha$ for scale $c = b - a$. Then:*

$$\mathbb{P}\left(\forall t \geqslant 1 : \left|\overline{Z}_t - \mu_t\right| < \frac{u\left(\sum_{i=1}^{t}\left(Z_i - \widehat{Z}_i\right)^2\right)}{t}\right) \geqslant 1 - 2\alpha.$$

**Theorem 10** (Partial statement of Audibert et al. [2], Theorem 1). *Let $X_1, \ldots, X_n$ be i.i.d. random variables bounded in $[0, s]$, for some $s > 0$. Let $\mu = \mathbb{E}[X_1]$ be their common expected value. Consider the empirical mean $\overline{X}_n$ and variance $V_n$ defined respectively by*

$$\overline{X}_n = \frac{\sum_{i=1}^{n} X_i}{n}, \quad and \quad V_n = \frac{\sum_{i=1}^{n}(X_i - \overline{X}_n)^2}{n}.$$

*Then for any $\delta \in (0, 1)$, with probability at least $1 - \delta$,*

$$\left|\overline{X}_n - \mu\right| \leqslant \sqrt{\frac{2V_n \log(3/\delta)}{n}} + \frac{3s \log(3/\delta)}{n}.$$

## F.2  Uniform-mass binning

Kumar et al. [21] defined well-balanced binning and showed that uniform mass-binning is well-balanced.

**Definition 6** (Well-balanced binning). A binning scheme $\mathcal{B}$ of size $B$ is $\beta$-well-balanced ($\beta \geqslant 1$) for some classifier $g$ if

$$\frac{1}{\beta B} \leqslant \mathbb{P}\left(g(X) \in I_b\right) \leqslant \frac{\beta}{B},$$

simultaneously for all $b \in [B]$.

To perform uniform-mass binning labeled examples are required at the stage of training the base classifier $g(\cdot)$. We denote this data as $\mathcal{D}_{\text{cal}}^1$. Procedures based on uniform-mass binning are well-balanced if $\left|\mathcal{D}_{\text{cal}}^1\right|$ is sufficiently large.

**Lemma 11** (Kumar et al. [21], Lemma 4.3). *For a universal constant $c > 0$, if $\left|\mathcal{D}_{cal}^1\right| \geqslant cB \ln(B/\alpha)$, then with probability at least $1 - \alpha$, the uniform mass binning scheme $\mathcal{B}$ is 2-well-balanced.*

The calibration guarantees in Section 4 depend on the minimum number of training points $N_{b^\star}$ in any bin. Uniform mass-binning guarantees that $N_{b^\star} = \Omega(n/B)$. This is used in the proof of Theorem 5.