[Reviews · NeurIPS 2020]

Review 1

Summary and Contributions: The paper focuses on three notions of uncertainty quantification: prediction sets, confidence intervals and calibration; for binary classification in the distribution-free setting. The authors establish a theorem that connects these three notions for score-based classifiers. They also show that distribution-free calibration is only possible using a scoring function whose level sets partition the feature space into at most countably many sets. Parametric calibration schemes do not satisfy this requirement, while nonparametric schemes based on binning do. They also derive distribution-free confidence intervals for binned probabilities for both fixed-width and uniform-mass binning. leading to distribution-free calibration.

Strengths: The results are interesting and relevant to the NeurIPS community because the quantifing uncertainty is important in safe AI. By unifying different three ways to quantify uncertainty, the paper provides the user a "unique" way of assessing uncertainty. Further, they discuss which post-hoc calibration techniques can guarantee calibrated uncertainty quantification and which not. In my opinion, the paper indirectly also shows that post-hoc calibration techniques should not be used, because they require |Dcal_2| to be proportional to n (if I understand the notation Ω(n), please define it), giving more reasons for using Bayesian nonparametric models (that directly estimate the distribution from the training data).

Weaknesses: I think that some of the examples that are in the appendix should have been inserted in the main paper to give to the reader an idea of the required size of the calibration sets. For instance, from appendix D.4, "Uniform mass binning was performed with 10 bins and 1940 data points from the source domain were used to estimate the quantiles. 7840 source data points were used for the calibration and finally, 28000 data points from the target domain were used for evaluation purposes. "

Correctness: Yes

Clarity: Yes

Relation to Prior Work: Yes, but I am not an expert on this topic and I am not sure if these results are fully novel.

Reproducibility: Yes

Additional Feedback:


Review 2

Summary and Contributions: This paper studies and connects three notions of prediction quality in a distribution-free setting (ie making no assumptions on the data distribution.) These include claibration, prediction sets, and confidence intervals, as well as their generalizations with respect to a hypothesis function f. The primary contribution of this work is to show that some level of discretization is necessary for distribution-free learning, even in infinite data.

Strengths: This paper is able to tell, from my perspective, a full story; they eloquently connect the notions of prediction uncertainty and then address some natural followup questions in Section 4. For example, they discuss additional assumptions needed for their results to hold under covariate shift, as well as guarantees that can be attained in the online setting.

Weaknesses: While there is lots of relevant work cited, the paper does not contextualize and differentiate its contribution clearly. While they reference the appendix for proofs, having a bit more intuition in the body of the paper would be helpful for some results (ex: Theorem 1.) Suggestions to improve score: - Spend more space contextualizing this work in the literature, and earlier in the paper. - Give intuition in body for *why* Theorems might be true. ================== AFTER RESPONSE ================== Thank you for the clarification on contextualization in the literature.

Correctness: From what I can tell, proofs are correct.

Clarity: Figure 1 was very helpful to understand the relationship between the results, and the paper as a whole was very well-organized and progressed naturally. One small thing to improve clarity (at least for some readers) is to make a note that while C can be any function, it can only be (1-alpha)-CI if it is measurable [equipped with the Borel sigma algebra]. This is just one example; while measurability never is explicitly mentioned, it is an underlying assumption of most of the paper. This is very understandable since most of ML takes this for granted, but a statistician or mathematician reading this might be confused. Take this with a grain of salt, since those readers are often secondary audience members in the NeurIPS community.

Relation to Prior Work: Mentioned in weaknesses, but I think (the clarity of) this is the paper's primary shortcoming at the moment. There is lots of prior work cited, but if a reader is unfamiliar with the cited results, it is hard to tell how this work distinguishes itself.

Reproducibility: Yes

Additional Feedback: Curious how this relates to the proper scoring rule literature (also rooted in Brier's seminal work.) See the following references that might be helpful: - Leonard Savage 1971: Elicitation of Personal Probabilities and Expectations JASA - Tilmann Gneiting Adrian Raftery 2007: Strictly Proper Scoring Rules, Prediction,and Estimation JASA


Review 3

Summary and Contributions: - This paper discusses three notions of uncertainty quantification: calibration, prediction sets (PS) and confidence intervals (CI). Focus is mainly on calibration. - The analysis specifically focuses on the distribution-free setting to attempt understand the kinds of possible valid uncertainty quantification without distributional assumptions on the data. - The paper connects the three notions above through theorems and proofs. - Strengthen recent observations that continuous output methods have unverifiable calibration errors, especifially in the distribution-free setting.

Strengths: - A well-written paper, despite that at times it can become hard to follow. - A strong theory paper with detailed discussions about theorems and proofs.

Weaknesses: - No experiments: The paper is densely packed with a lot of theory though there is not a single experiment in the entire paper. Had this been a groundbreaking or new work looking into the ability of reliably estimating the calibration of continuous output methods, then this would have fallen under a minor issue. But seeing as much of the work has previously already been shown (i.e. Histogram binning is needed for verifiable calibration estimation [21 in paper]), I feel some experiments are missing. That being said, it is still a good contribution but showing some of the sort experiments is still needed. Maybe such a paper is better suited for conferences which have a heavy focus on theory. - Some sort of toy examples to convey the theory better would also strengthen the paper. Currently, there is a lot of (useful) theory, though it is hard to grasp without examples or visualizations. - Line 241 the authors state: "which would make the pi\_b estimates less calibrated": What exactly is meant here? Using fixed-width binning the bins do not get enough samples, though this causes a high variance approximation of the empirical estimate. Therefore, the problem with too few samples lies in the insufficiency of samples for empirical estimation. So I do not get what is meant by the estimate is "less calibrated". You can still be perfectly calibrated (even when samples rarely fall into a specific bin) regardless of the sample set size, the issue lies in the fact that it is hard and often impossible to measure the calibration for this bin given the limitation of the datasets. So I do not even get how this kind of statement can be made by stating the estimate pi\_b is less calibrated. Maybe I have misunderstood something. I am curious to hear feedback regarding this from the authors. - Much of the discussion about histogram binning seems to be heavily based on previous works. In the "summary", it is claimed that: "In contrast, we showed that a standard nonparametric method – histogram binning – satisfies approximate and asymptotic calibration guarantees without distributional assumptions." I feel this is a strong statement (e.g. without reference to other works) as this has been partially shown before it other works (which have been cited in this work earlier). - Similar comment regarding the takeaway message: recommendation of some form of binning as been done before already to obtain verifiable calibration errors. - At the time it is hard to follow as the theory is very dense. - In summary, the paper presents a different and new perspective at calibration, though the core idea as already been presented before. Especially, the take away from this paper has already been presented before.

Correctness: claims: Yes, though it is a densely packed paper so it is hard to verify everything. method: No. There is no method presented. Nothing to take away.

Clarity: Yes, though could be improved by introducing more figures/illustrations instead of 8 pages filled with dense theory.

Relation to Prior Work: Yes. Though at times observations are phrased as if this paper introduced them where they have been introduced before.

Reproducibility: No

Additional Feedback: There is not a single "result" presented in this paper.

[Author Response · NeurIPS 2020]

We thank the reviewers for their constructive feedback. Our responses follow. We use DF for 'distribution-free'.

**[R1]** **'...if I understand the notation $\Omega(\mathbf{n})$...'** The interpretation of $\Omega(n)$ is correct; we would like to point out that
$n$ is #calibration points, which only depends on the desired calibration error $\epsilon$ (typically, $n \approx 1/\epsilon^2$), and does not
have to be a constant fraction of #training points. As far as we know, Bayesian calibration does require distributional
assumptions; examining whether they yield DF calibration is an interesting direction for future work.

**'...examples that are in the appendix...move to the main paper...'** We agree. If the paper is accepted, we will add
explicit numbers and examples using the extra ninth page.

**[R2]** **'...the paper does not contextualize and differentiate its contribution clearly...'** The literature on calibration
can be split in two parts: how to *measure* calibration (Brier score, ECE error, proper scoring rules, etc) and how to
*achieve* calibration (Platt scaling, binning, etc). Our work falls in the second category. Within this, the only DF results
we are aware of stem from "Venn prediction", which we discuss in Appendix E (however, their theoretical approach
does not lead to an actionable algorithm with a DF guarantee). If the paper is accepted, we will contextualize our work
as above (including its history in meteorology/statistics) in more detail using the extra ninth page.

**'...more intuition for Theorem 1...'** We agree. On line 127 we have added the following

$$\underbrace{|\mathbb{E}\left[Y \mid f(x)\right] - f(x)| \leqslant \epsilon(f(x))}_{\text{calibration}} \implies \underbrace{f(x) \in C(f(x))}_{\text{CI wrt } f} := [f(x) - \varepsilon(f(x)), f(x) + \varepsilon(f(x))],$$

and on line 128, we have called $u, l, m$ as the left-endpoint, right-endpoint and midpoint functions respectively and $\varepsilon$ as
the constant function returning the largest interval radius. We also have intuition to share about Theorem 2; we have
now added a paragraph just after describing how, and why, the proof works (not reproduced here due to limited space).

**'...function C can only be (1-alpha)-CI if it is measurable...'** We agree. Measurability is indeed required. We have
now added a clarification/remark in the notations paragraph.

**[R3]** **'...[21] previously showed histogram binning is needed for calibration... this has been partially shown**
**before it other works (sic)...'** We respectfully disagree, as we explain next. [21] does make claims about the difficulty
of *evaluating/measuring* miscalibration of Platt/temperature scaling, but these are not formal impossibility results.
Mathematically, [21, Thm 4.1] showed that binning *suffices* for calibration, and compares scaling+binning to the best
within a fixed, regular, injective parametric class. In contrast, our results are new in the DF setting, showing not only
that binning is necessary for calibration (Thm 3), but also sufficient (Cor 4). Thm 3 in particular implies that the best in
the aforementioned parametric class in [21] itself cannot have a DF guarantee, and it also yields a formal impossibility
result for Platt/temperature scaling. We will clarify this in our expanded discussion about related work.

**'...not a single experiment...'** Appendix D has 'proof-of-concept' simulation under covariate shift, but we agree that
the focus of this paper is theoretical, specifically to shed light on what is or is not achievable without making (usually
unverifiable) distributional assumptions. Luckily, our theory applies to a large swatch of popular existing methods, for
which a lot of empirical work already exists in the literature.

**'...a lot of (useful) theory, though it is hard to grasp without examples or visualizations...'** We do have intu-
ition/examples to share. If accepted, we will use the extra ninth page to add these.

**'Line 241 "which would make the pi_b estimates less calibrated"... What exactly is meant here?...'** We agree the
current phrasing may be misleading. We intended to say something straightforward: with fixed-width binning, some
bins may have much fewer samples relative to others, and calibration cannot be guaranteed/verified for those bins. This
is the typical motivation for uniform-mass binning. We have updated the text to clarify this with a clearer word choice.

[Meta-Review · NeurIPS 2020]

This is a theory-focused paper that analyzes the relations between prediction sets, confidence intervals and calibration in a distribution-free setting. Reviewers were unanimous in supporting the acceptance of this paper to the conference. Since the topic has been extensively researched before they had some difficulty teasing apart the unique contributions of this paper. Authors have promised to clarify this in the final version. Reviewers were also keen on seeing better illustration and description of the intuitions behind the theorems. That would help the paper have a wider impact in the community.